# Blocking phospholamban with VHH intrabodies enhances contractility and relaxation in heart failure

Erwin De Genst [1,2 ✉], Kylie S. Foo[2,3], Yao Xiao[2,3], Eduarde Rohner [2,3], Emma de Vries[1], Jesper Sohlmér [2,3], Nevin Witman [2], Alejandro Hidalgo[2,4], Terje R. S. Kolstad[5,6], William E. Louch[5,6], Susanne Pehrsson[4], Andrew Park[7], Yasuhiro Ikeda[7], Xidan Li[2], Lorenz M. Mayr[1,8], Kate Wickson[1], Karin Jennbacken [4], Kenny Hansson[4], Regina Fritsche-Danielson [4], James Hunt [1 ✉] & Kenneth R. Chien [2,3 ✉]

The dysregulated physical interaction between two intracellular membrane proteins, the sarco/endoplasmic reticulum $Ca^{2+}$ ATPase and its reversible inhibitor phospholamban, induces heart failure by inhibiting calcium cycling. While phospholamban is a bona-fide therapeutic target, approaches to selectively inhibit this protein remain elusive. Here, we report the in vivo application of intracellular acting antibodies (intrabodies), derived from the variable domain of camelid heavy-chain antibodies, to modulate the function of phospholamban. Using a synthetic VHH phage-display library, we identify intrabodies with high affinity and specificity for different conformational states of phospholamban. Rapid phenotypic screening, via modified mRNA transfection of primary cells and tissue, efficiently identifies the intrabody with most desirable features. Adeno-associated virus mediated delivery of this intrabody results in improvement of cardiac performance in a murine heart failure model. Our strategy for generating intrabodies to investigate cardiac disease combined with modified mRNA and adeno-associated virus screening could reveal unique future therapeutic opportunities.

[1] Discovery Sciences, R&D, AstraZeneca, Cambridge, UK. [2] Karolinska Institutet/AstraZeneca Integrated Cardio Metabolic Centre (KI/AZ ICMC), Department of Medicine, Karolinska Institutet, Huddinge, Sweden. [3] Department of Cell and Molecular Biology, Karolinska Institutet, Stockholm, Sweden. [4] Bioscience Cardiovascular, Research and Early Development, Cardiovascular, Renal and Metabolism (CVRM), BioPharmaceuticals R&D, AstraZeneca, Gothenburg, Sweden. [5] Institute for Experimental Medical Research, Oslo University Hospital and University of Oslo, Oslo, Norway. [6] K.G. Jebsen Centre for Cardiac Research, University of Oslo, Oslo, Norway. [7] Biologics Engineering, R&D, AstraZeneca, One MedImmune Way, Gaithersburg, MD, USA. [8] Present address: Vector BioPharma AG, Aeschenvorstadt 36, 4051 Basel, Switzerland. ✉email: erwin.degenst@astrazeneca.com; james.hunt1@astrazeneca.com; kenneth.chien@ki.se

Abnormal intracellular protein–protein interactions cause aberrant signalling processes in a range of major diseases, including heart failure, cancer and neurodegenerative disorders[1]. Heart failure is characterised by defective $Ca^{2+}$ signalling, which results in impaired cardiac muscle contraction and relaxation as well as the development of cardiac arrhythmias and adverse remodelling[2–4]. An important contributor to this pathology is the reduced activity of the sarco/endoplasmic reticulum $Ca^{2+}$-ATPase (SERCA2a) and increased relative levels of its inhibitor, non-phosphorylated phospholamban (PLN)[5]. In addition, several hereditary pathological variants of PLN have been directly linked to dilated cardiomyopathy, including the variants R9C[6,7], R9L/H[8], R25C[9], R14del[10], and the truncation mutant L39X[11]. Strategies to enhance calcium cycling by increasing SERCA2a activity using gene therapy[12,13] or inhibiting PLN function are therefore very attractive therapeutic approaches for heart failure[14–16]. However, no small molecule drug screening approach to-date has successfully identified a lead candidate. This is partly due to the fact that PLN is an integral intracellular membrane protein with flexible domains and lacking defined and accessible druggable pockets. Furthermore, PLN exists in a dynamic equilibrium between different structural forms, including monomeric and pentameric states, and complexes with modulator proteins, perpetually regulating its inhibitory SERCA2a function[17,18]. PLN function is further controlled by phosphorylation at two different sites, Ser 16 and Thr 17, through Protein Kinase A (PKA) and $Ca^{2+}$ calmodulin-dependent Protein Kinase (CamKII), respectively, and by dephosphorylation by protein phosphatase I (PPI)[19]. This highly dynamic interchange of structural-functional states of PLN considerably complicates small molecule screening approaches and renders it very challenging to identify compounds with high enough potency to elicit desired functional changes.

A promising route to target intracellular protein–protein interactions, such as the PLN:SERCA2a interaction, is through the use of intrabodies. Intrabodies are antibody-like molecules that can fold, and recognise their binding partner, within the cell[20]. These molecules are increasingly utilised as valuable tools to study the function of intracellular proteins[21,22] and have been successfully used to modify intracellular protein–protein interactions for targets involved in cancer and neurodegeneration[23,24]. Furthermore, intrabodies can rapidly modulate the function of a target directly at the protein level, as well as specifically recognise conformational or post-translational states[22]. These features offer significant advantages over genetic knock-down methods such as antisense oligonucleotides (ASO), CRISPR or siRNA, which require efficient turnover and which lack the specificity for different post-translational functional states and conformations of the target.

The VHH domains of camelid heavy-chain antibodies[25–27] show particular promise for their use as intrabodies[28]. VHHs are composed of a single highly soluble domain, a tenth the size of a full-length conventional antibody, with identical affinities and specificities to those of the intact antibody molecule. VHHs also tend to maintain their binding properties within the cytosol[27,29]. Furthermore, the amino-acid sequence of the VHH framework is closely related to human VH sequences[30] and is easily humanised to compensate for any unwanted immunogenicity effects in clinical applications[31,32].

Contrary to small molecules, intracellular delivery has long been a major hurdle for protein-based therapeutic strategies. However, advances in nucleic acid-based delivery approaches, such as nucleic acid loaded nanoparticles[33] or DNA viral methods[34], have led to several FDA and EMA approved gene therapies in recent years[35,36]. Chemically modified messenger RNA (modRNA)[37–39], which are synthetic messenger RNA molecules that can act as a drug to replace or introduce in vivo therapeutic proteins, are rapidly emerging as viable gene therapy modalities. ModRNAs showcase enhanced stability and low immunogenicity, while without the potential safety risks of genome integration often seen with DNA based vectors[38,39]. Both DNA- and RNA-based technologies are actively being pursued for the purpose of sustained delivery of protein-based therapeutics. As such, these in vivo expressed biologics (IVEB) platforms present an opportunity for intrabodies.

Here we report a rapid, robust and efficient pipeline for the development of intrabody based IVEB, by a combination of simple in vitro selection of a synthetic VHH phage-display library for intrabody discovery and the use of small modRNA libraries for rapid intracellular and in vivo validation, followed by the in vivo delivery of intrabodies using AAV vectors for validation of physiological effects. We applied our method to PLN as a case study. By developing candidate VHH intrabodies, using in vitro selection techniques and straightforward protein engineering, we were able to rapidly investigate different mechanisms to interfere with PLN function, to finally deliver a unique molecule capable of in vivo activity in a murine heart failure model.

## Results

**Phage-display selection and VHH engineering identify high-affinity VHHs specific for different functional states of PLN.** PLN is a highly conserved intracellular transmembrane protein with only the regulatory region, containing the phosphorylation sites Ser 16 and Thr 17 and a small hydrophilic linker region of PLN exposed to the cytosol (Fig. 1a)[17]. To obtain VHHs that can bind to PLN, we chose to interrogate a fully synthetic VHH phage-display library in vitro with peptides that have sequences corresponding to those located within the cytoplasmic regulatory domain of PLN (Fig. 1a, Table 1, online methods). Both non-phosphorylated peptides and phosphorylated peptides at residue Ser 16 (the most abundant phosphorylated form of PLN) were immobilised on streptavidin-coated magnetic beads and used to perform several rounds of bead-based selection.

After three rounds of selection, single *E. coli* phage infected colonies were picked and grown for VHH expression in small-scale cultures and screened for PLN or pS16PLN binding by ELISA. This resulted in the identification of VHH B4, which binds to both the non-phosphorylated PLN and pS16 PLN peptides; and VHH C6, which is specific for p16S PLN peptides. Small-scale protein purification of these VHHs was then performed to measure the binding rate constants ($k_a$ and $k_d$) and dissociation constant ($K_d$) of both VHHs for binding to PLN and pS16 PLN peptides using surface plasmon resonance (SPR). The $K_d$ value was found to be 44 nM for the VHH B4 interaction with non-phosphorylated PLN peptides and 682 nM for the VHH B4:pS16 peptide interaction (Fig. 1b, Supplementary Table 1). The $K_d$ value of the VHH C6:pS16 PLN interaction was found to be equal to 680 nM. No detectable binding was observed for VHH C6 with non-phosphorylated PLN (Fig. 1b), confirming the specificity of this VHH for the pS16 PLN peptide.

In order to test if we can increase the binding potency of the VHHs, we constructed a bivalent variant of VHH B4, $VHH_2$ B4B4. We also reasoned that due to the avidity effect, the binding of the $VHH_2$ would be biased for interaction with PLN oligomers, which could offer strong selectivity towards the functionally relevant pentameric state of PLN. SPR experiments revealed that the apparent $K_d$ value of $VHH_2$ B4B4 for binding to PLN or pS16 PLN was equal to 0.15 and 1 nM, respectively, thus resulting in a ~300-fold (PLN) and ~600-fold (pS16 PLN)

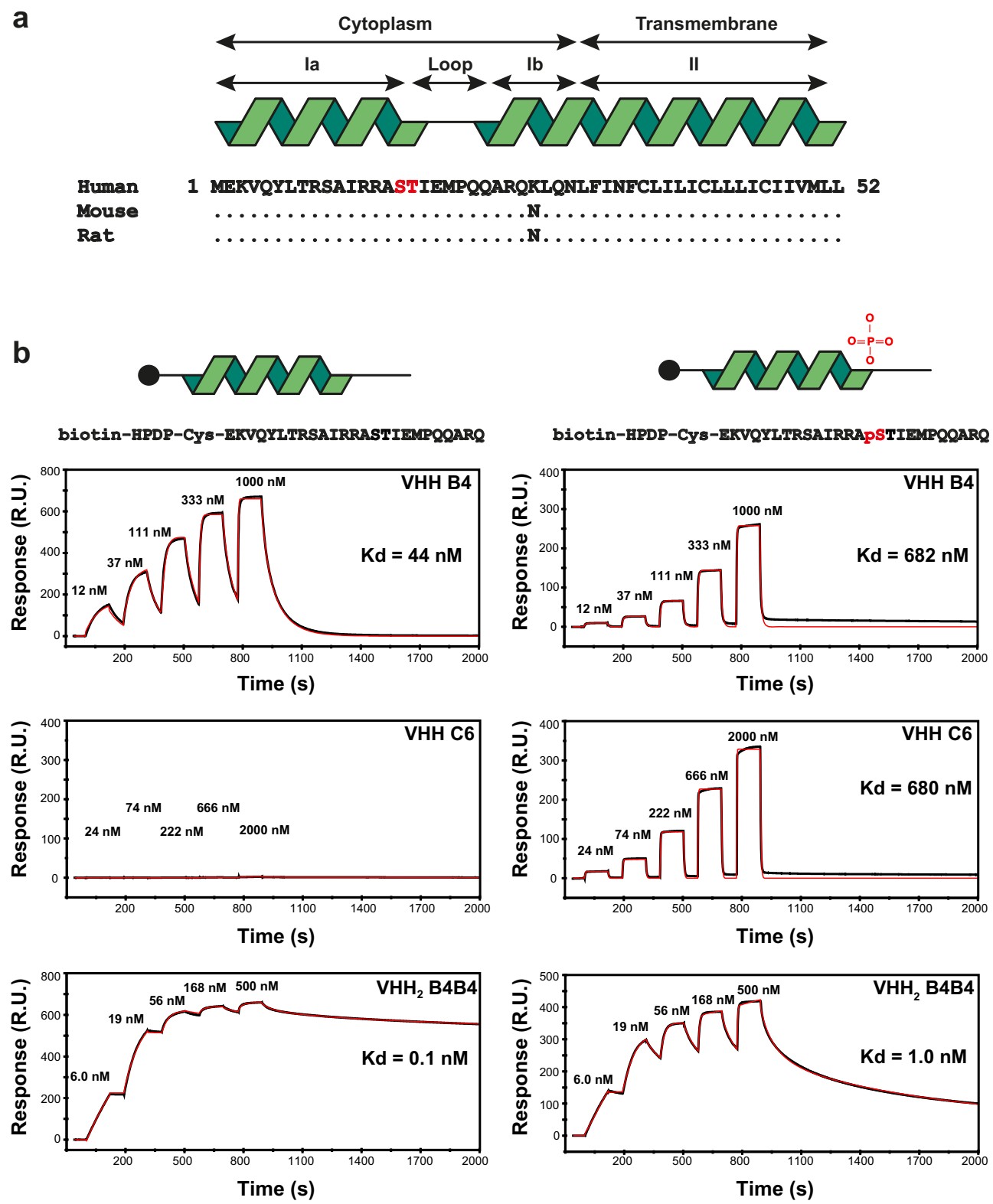

increase in binding potency compared to the monomeric VHH B4 binding (Fig. 1b).

In summary, two potent VHHs with different specificity for the post-translational state of PLN were selected from a synthetic VHH library. VHH dimerisation significantly increased the binding potency for PLN and pS16 PLN dramatically due to avidity, which offers potential selectivity towards oligomeric states of PLN.

**Intrabody modRNA transfection results in high expression levels and demonstrates intracellular target engagement.** Following the successful selection and biophysical characterisation of VHH B4, VHH C6 and the bivalent variant VHH$_2$ B4B4, we next evaluated their intracellular expression and binding to endogenous PLN by transfecting cell lines with modRNA molecules that code for the different intrabody variants. The intrabodies were expressed as hemagglutinin (HA) tag (Fig. 2a) or HA-mCherry

**Fig. 1 PLN sequence alignment and surface plasmon resonance measurements of the VHH and $VHH_2$ interaction with PLN peptides. a** PLN primary sequence and secondary structure annotation: ribbon, helix; line, random coil. PLN domain names and their subcellular localisation are annotated. The phosphorylation sites in the sequence, Ser 16 and Thr 17, are indicated in red. The amino-acid sequence of human PLN is aligned with the sequences of mouse and rat PLN and the amino-acid differences are indicated. **b** Surface plasmon resonance single-cycle kinetic traces of VHH B4, VHH C6 and $VHH_2$ B4B4 binding to the immobilised peptides biotin-HPDP-Cys-EKVQYLTRSAIRRASTIEMPQQARQ and biotin-HPDP-Cys-EKVQYLTRSAIRR ApSTIEMPQQARQ (biotin-HPDP-Cys moiety represented as a black dot) at different concentrations of the VHH or $VHH_2$. The panels on the left show the binding data for the VHH/$VHH_2$ interaction with the non-phosphorylated peptide and the panels on the right represent the binding data for the VHH/ $VHH_2$ pS16:p16S PLN peptide interactions. The measured concentrations of the VHHs are indicated above each association trace and the $K_d$ values of the interaction of the VHH or $VHH_2$ with PLN or pS16 PLN are noted in the graph.

| Name | Sequence[a] |
|---|---|
| I | Cys-EKVQYLTRSAIRRASTIE-NH2 |
| II | Cys-EKVQYLTRSAIRRASTIEMPQQARQ-NH2 |
| III | Cys-EKVQYLTRSAIRRApSTIEMPQQARQ-NH2 |
| IV | biotin-EKVQYLTRSAIRRASTIE-NH2 |
| V | biotin-HPDP-Cys-EKVQYLTRSAIRRASTIEMPQQARQ-NH2 |
| VI | biotin-HPDP-Cys-EKVQYLTRSAIRRApSTIEMPQQARQ-NH2 |

**Table 1 Amino-acid sequences of peptides containing PLN residues used for VHH phage-display selections, screening and SPR experiments.**

[a]PLN amino-acid sequences are shown in single letter code. Peptide modifications include: *Cys* cysteine residue, *NH2* C-terminal amidation, *biotin* amino terminal linked biotin-group; biotin-HPDP-Cys, N - [6 - (Biotinamido) hexyl] - 3′ - (2′-pyridyldithio) propionamide (EZ link HPDP-Biotin) Cysteine persulphide.

fusions (Supplementary Fig. 1a) at the C-terminus of the intrabody sequence. High transfection efficiencies and intracellular expression were confirmed in multiple cell models, including HeLa cells, hESC, iPSC cardiomyocytes and rat neonatal cardiomyocytes (Supplementary Table 2 and Supplementary Figs. 1-3). Expression was visible after 4 h and peaked after 10–15 h of transfection (Supplementary Figs. 1b and 2), followed by a slow decline, typical for the transient nature of modRNA transfection[37]. Transfections of modRNA-encoded intrabodies in different formats, including, monomeric, bivalent or genetic fusions with fluorescent proteins, showed furthermore no detectable cytotoxicity (Supplementary Fig. 4a, b).

To evaluate if the intrabodies recognise the endogenously expressed PLN in primary cells, we transfected the modRNA-encoded intrabodies in freshly isolated and cultured rat neonatal cardiomyocytes and performed co-immunoprecipitation experiments. Immunoprecipitation of both intrabodies VHH B4 and $VHH_2$ B4B4 using an anti-HA antibody demonstrated the presence of PLN bound intrabody within the cell lysates (Fig. 2b). Furthermore, we confirmed the selectivity of $VHH_2$ B4B4 for pentameric PLN conformers by titrating increasing amounts of $VHH_2$ B4B4 modRNA for cell transfection and using extraction buffers that allow efficient extraction of the pentameric PLN species (RIPA buffer) and bead elution conditions that preserve the native conformations of PLN (Fig. 2c). The resulting western blot analysis showed a clear enrichment of pentameric PLN fraction after the co-IP experiment, with even higher enrichment at lower stoichiometries $VHH_2$ B4B4/PLN (Fig. 2c), consistent with the avidity-based binding mechanism. For the VHH C6 intrabody, although expression levels were similar to the VHH B4 and $VHH_2$ B4B4, we could not confirm target engagement for this intrabody using co-IP studies, possibly due to a lower fraction of pS16 PLN compared to non-phosphorylated PLN and extraction conditions that do not preserve binding. Immunofluorescence imaging on fixed rat neonatal cardiomyocytes, however, showed co-localisation of VHH C6 with PLN (Fig. 3i).

Collectively, we have demonstrated that the VHH identified can be highly expressed in mammalian cell lines and primary rat neonatal cardiomyocytes using modRNA transfection, and that these intrabodies are able to engage with their endogenously expressed targets.

**Calcium cycling in rat neonatal cardiomyocytes reveal functional differences between intrabodies with different PLN specificity.** To understand the functional effects of the intrabodies on the calcium handling of live cardiomyocytes, we isolated and cultured neonatal cardiomyocytes from 6-day-old rat pups and transfected these cells with the modRNA constructs shown in Fig. 2. Calcium flux imaging of transfected cells, using Fluo-4 AM loading and electrical stimulation (Fig. 3a-d), demonstrated that the $T_{50}$ values of the $Ca^{2+}$ transients of cells transfected with VHH B4, and $VHH_2$ B4B4 modRNA were significantly lower than the untransfected control cells ($P < 0.0001$) (Fig. 3e and Supplementary Table 3), indicative of enhanced calcium reabsorption. A smaller, but statistically significant decrease of the $T_{50}$ value compared to control cells was also observed for cells transfected with VHH C6 modRNA ($P = 0.0012$) (Fig. 3e and Supplementary Table 3). In addition, the $Ca^{2+}$ decay rates, $1/\tau$, were observed to be significantly higher in cells transfected with VHH B4 and $VHH_2$ B4B4 modRNA, compared to control cells ($P < 0.0001$) (Fig. 3f and Supplementary Table 3). In contrast, no significant differences in the $1/\tau$ values were detected between control cells and VHH C6 modRNA transfected cells (Fig. 3f and Supplementary Table 3). Additionally, we observed significant increases in upstroke velocity for cells transfected with VHH B4 and $VHH_2$ B4B4 modRNA compared to untransfected cells ($P < 0.0001$) (Fig. 3g, Supplementary Table 3). For VHH C6 modRNA transfected cells we did not observe any statistical differences in upstroke velocity ($dF/dt_{up}$) compared to untransfected cells. The $Ca^{2+}$ flux measurements further show a reduction in the average peak amplitude ($\Delta F/F_{0\ max}$) of the $Ca^{2+}$ transient for VHH C6 modRNA transfected cells compared to control cells ($P < 0.0001$) (Fig. 3h, Supplementary Table 3), whereas the average peak amplitudes for the VHH B4 and $VHH_2$ B4B4 transfected cells were not found to be significantly different to that of the control cells (Fig. 3h, Supplementary Table 3).

Immunofluorescence staining of fixed cells using anti-HA and anti-PLN antibodies confirmed high transfection efficiencies for all the intrabodies. For both HA and PLN, stronger staining intensity was observed in the perinuclear region of the cells compared to the cytosol, suggesting that the intrabodies are mainly localised to the sarcoplasmic reticulum and co-localised with PLN (Fig. 3i). For cells transfected with the intrabody $VHH_2$ B4B4, a large number were only weakly stained using the PLN antibody. This is consistent with the fact that the epitopes of the intrabody and the anti-PLN antibody, 2D12, overlap; due to strong and bivalent binding, enable $VHH_2$ B4B4 molecules to render this epitope inaccessible for the primary anti-PLN antibody 2D12 (Fig. 3i).

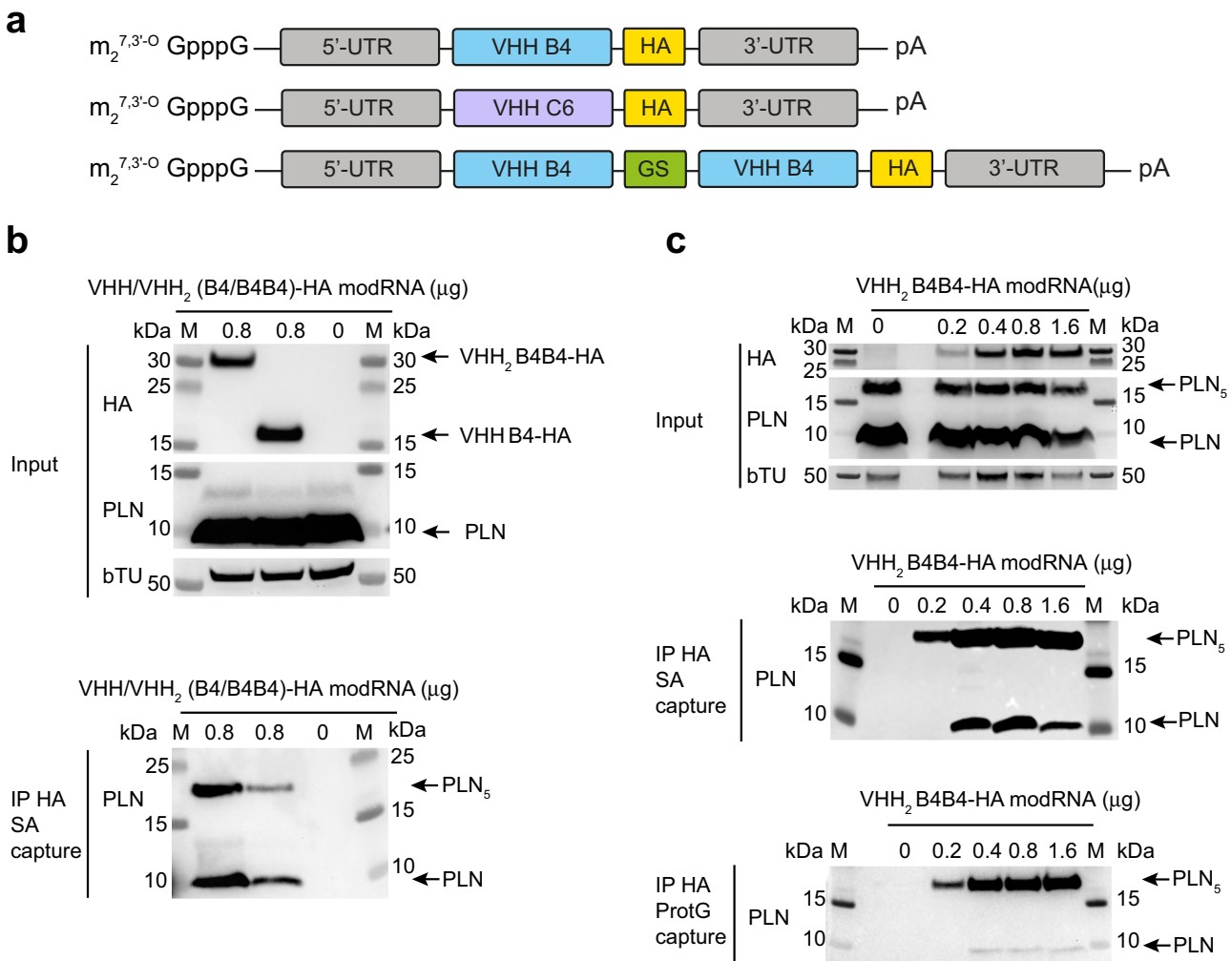

**Fig. 2 Endogenous target engagement of intrabodies in rat neonatal cardiomyocytes using modRNA transfection. a** modRNA construct design. $m_2^{7,3'-O}$ GpppG is the ARCA CAP; 3'UTR is the 3'untranslated region, VHH is the intrabody sequence, GS is a sequence that encodes a glycine-serine linker peptide (GGGSGGGSGGGSGGGS), HA is the hemagglutinin tag, 5' UTR is the 5' untranslated region and pA is the poly-A tail. **b** Co-immunoprecipitation experiments using lysates of VHH B4-HA and $VHH_2$ B4B4-HA modRNA transfected rat neonatal cardiomyocytes. Molecular weight marker is indicated (M) and the sizes of the marker bands in kDa are noted at either side of the blot. Arrows and labels on the right indicate the size of different molecular species of PLN (PLN and $PLN_5$) or intrabody (VHH B4 and $VHH_2$ B4B4). Top panel: Western blots of cell lysates (boiled and reduced), probed with anti-HA, anti-PLN and anti-β-tubulin antibodies (left). Lower panel: Western blot of streptavidin magnetic bead-based co-IP using native elution and SDS-PAGE conditions and PLN detection. More than three independent Co-IP experiments successfully replicated the findings. **c** Co-immunoprecipitation experiments using lysates of $VHH_2$ B4B4 modRNA transfected rat neonatal cardiomyocytes at different concentrations of modRNA. Molecular weight marker is indicated as M and sizes of the bands in kDa noted at either side of the blot. Arrows and labels indicate the size of different molecular species of PLN (PLN and $PLN_5$). Top panel: Western blots of input samples (non-boiled and non-reduced), with anti-HA, anti-PLN and anti-β-tubulin antibodies (left). Middle panel: Western blot of streptavidin magnetic bead-based co-IP using native elution and SDS-PAGE conditions and anti-PLN detection. Bottom panel: Western blot of protein-G magnetic bead-based co-IP using native elution and SDS-PAGE conditions and anti-PLN. This concentration-dependent co-IP experiment was not independently repeated. Source data are provided as a Source Data file.

With the $Ca^{2+}$ imaging experiments, we showed remarkable disparities in the functional cellular phenotype between the intrabodies on the $Ca^{2+}$ transient when expressed in rat neonatal cardiomyocytes. VHH B4 and to a greater extent, $VHH_2$ B4B4 showed enhanced lusitropic and inotropic effects. Alternatively, VHH C6 had smaller or opposite effects to those of the other two intrabodies. Immunofluorescent staining of the cardiomyocytes further confirmed the co-localisation of the intrabodies with PLN, consistent with target engagement.

**In vivo delivery of modRNA-encoded intrabodies reveal significant functional effects in mature cardiomyocytes.** Specific target engagement of the intrabodies and observed effects on

calcium cycling in cultured rat neonatal cardiomyocytes prompted us to investigate the potential to deliver the modRNA-encoded intrabodies in vivo to determine their effects on healthy mature cardiomyocytes. For these experiments, we focused on the effects of the $VHH_2$ intrabody B4B4 and for a comparable assessment of pS16PLN targeting to include a new modRNA construct encoding for a bivalent version of VHH C6, $VHH_2$ C6C6. To this end, we first evaluated the expression levels and tissue distribution of the intrabodies in WT C57BL/6 mice at 24 h and at 72 h after intra-myocardial injection of the modRNA (Fig. 4).

Twenty-four hours post-injection, western blots of the heart lysates showed high expression levels for $VHH_2$ B4B4 and lower levels for the $VHH_2$ C6C6. After 72 h, a significant expression level reduction was seen for both intrabodies (Fig. 4b, c),

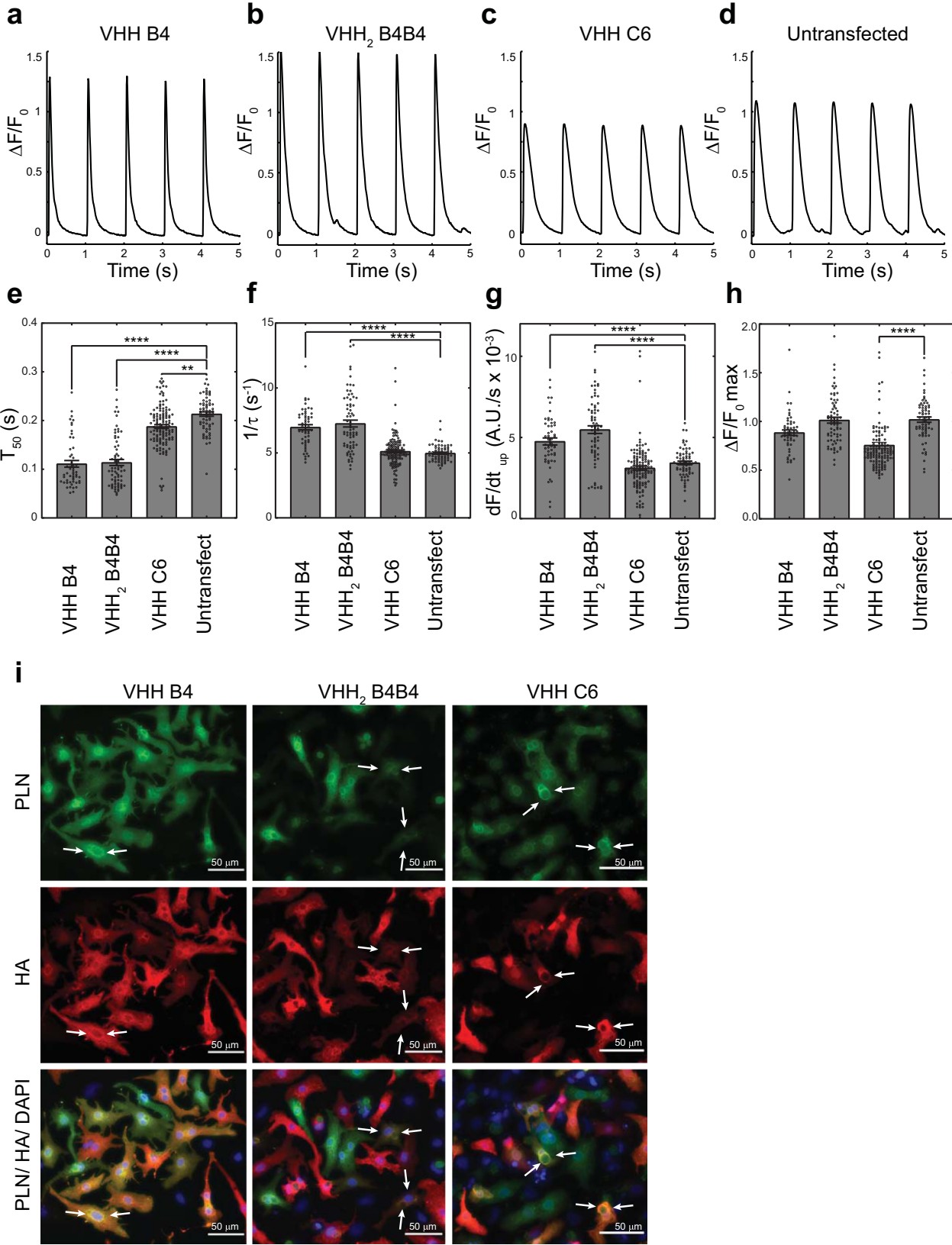

consistent with a typical transient expression observed for modRNA transfection[37]. Positively stained cardiomyocytes also showed a striated staining pattern coincident with the staining pattern produced by anti-SERCA2a antibody staining (Fig. 4d), indicating that both intrabodies and SERCA2a are co-localised at the sarcoplasmic reticulum.

Following the expression study, we evaluated the effects of the intrabodies on the calcium cycling in adult cardiomyocytes using modRNA delivery. Using Langendorff isolation, we collected cardiomyocytes from three different WT C57BL/6 mice that had been treated for 24 h with modRNA coding for T2A-mCherry genetic fusions of VHH$_2$ B4B4 and VHH$_2$ C6C6. For these Ca$^{2+}$

**Fig. 3 Effects of the modRNA-encoded intrabodies on the Ca$^{2+}$ dynamics of rat neonatal cardiomyocytes.** Top panels show representative Ca$^{2+}$ transients for the different groups: **a** cells transfected with modRNA encoding VHH B4 ($n = 53$); **b** cells transfected with modRNA encoding VHH$_2$ B4B4 ($n = 70$); **c** cells transfected with modRNA encoding VHH C6 ($n = 129$); **d** Untransfected control cells ($n = 66$). **e–h** Individual data points (open circles) and average value (bar graph) of **e** $T_{50}$ (peak half-width), **f** $1/\tau$ (rate of transient decay), **g** $dF/dt_{up}$ (the upstroke velocity) and **h** $\Delta F/F_{0\ max}$ (the amplitude of the Ca$^{2+}$ transient) for the control and treated cells with modRNA encoding for the different intrabodies (indicated on the X-axis): VHH B4 ($n = 53$); VHH C6 ($n = 129$); VHH B4B4 ($n = 70$); and untransfected control cells ($n = 66$). Error bars represent the standard error of the mean (SEM) in the sample set. Two data points in the VHH C6 Ca$^{2+}$ transient amplitude data with values above 2.0 were omitted from the graph in panel **h** for clarity. Statistical significant differences with the control group are indicated and were evaluated using one-way ANOVA using all data points and assuming normal distributions. Two-tailed post-hoc Tukey-Kramer $P$-values were calculated, and significance values for differences with the control group are indicated: **$P$-value < 0.01, ****$P$-value < 0.0001. Exact $P$-values are given in Supplementary Table 3. **i** Post-calcium imaging immunofluoresence images of fixed and permeabilised transfected cardiomyocytes (anti-PLN, green; anti-HA tag, red). White arrows indicate cells showing clear co-localisation and perinuclear staining for both the anti-HA antibody recognising the expressed intrabody and the staining originating from the 2D12 antibody for PLN staining. Immunofluorescence experiments of transfected rat neonatal cardiomyocytes have not been independently repeated. A.U. arbitrary units. Source data are provided as a Source Data file.

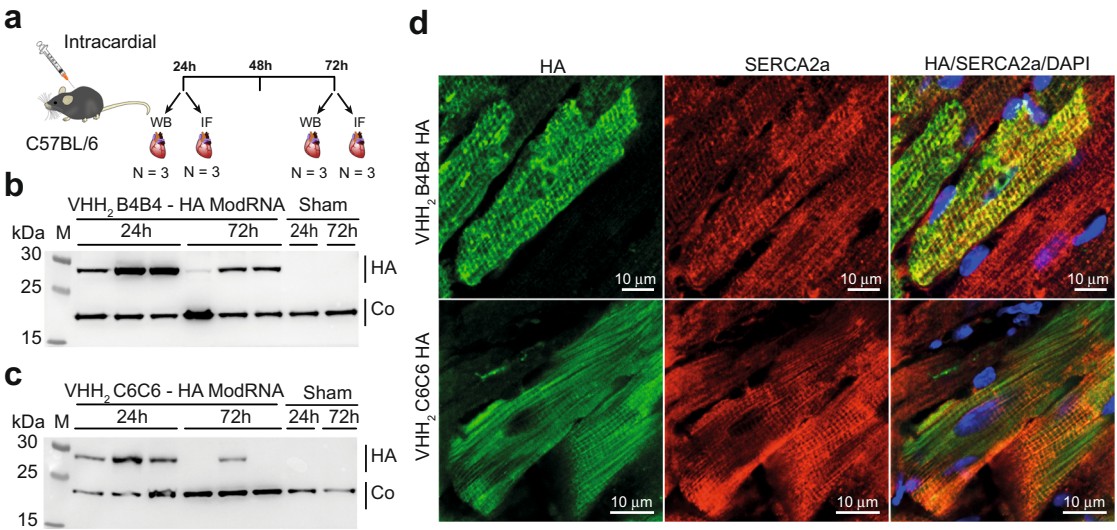

**Fig. 4 In vivo expression and tissue distribution of intramyocardial injected modRNA of the intrabodies VHH$_2$ B4B4-HA and VHH$_2$ C6C6-HA.** **a** Overview of the experimental procedure: Mice ($n = 12$) were injected intramyocardially with modRNA, sacrificed after 24 or 72 h; hearts were collected for western blot analysis ($n = 3$ at 24 h, $n = 3$ at 72 h); and immunofluorescence imaging (IF) of tissue sections ($n = 3$ at 24 h, $n = 3$ at 72 h). 6 mice, not treated with modRNA were used as control samples: western blot ($n = 2$ at 24 h; $n = 2$ at 72 h) and IF ($n = 1$ at 24 h; $n = 1$ at 72 h) **b**, **c** Western blot analysis of whole heart lysates (VHH$_2$ B4B4: $n = 3$ at 24 h, $n = 3$ at 72 h; VHH$_2$ C6C6-HA: $n = 3$ at 24 h, $n = 3$ at 72 h; control: $n = 1$ at 24 h, $n = 1$ at 72 h). The membranes were stained with anti-HA to reveal the expression of the HA tagged intrabodies: **b** VHH$_2$ B4B4-HA, **c** VHH$_2$ C6C6-HA. An anti-cofilin antibody was used as a loading control. Lane 1 contains a molecular weight marker (M) and sizes of the bands (in kDa) are indicated on the left. Western blots were performed $n = 3$ times with similar results. **d** Immunofluorescence imaging of transfected mouse hearts after 24 h of in vivo expression of VHH$_2$ B4B4-HA (top panels) and VHH$_2$ C6C6-HA (bottom panels). Slices were stained with DAPI, the SERCA2a antibody (red) and the anti-HA antibody (green). Identical immunofluorescence staining was repeated twice for tissue sections of mice transfected with VHH$_2$ B4B4-HA at the 24 h time point and repeated once for tissue sections of mice transfected with VHH$_2$ C6C6-HA at the 24 h time point (using 10 slices per heart) and resulted in similar staining patterns. Source data are provided as a Source Data file.

measurements, we also included a bivalent VHH intrabody, VHH$_2$ Cas9 (Fig. 5a) with specificity for CRISPR associated protein 9 (Cas9), as a non-relevant intrabody control. The presence of a T2A sequence[40,41] between the intrabody coding region and that of the mCherry reporter allows their translation as two separate proteins from a single messenger RNA molecule. This permits the detection of intrabody-expressing cells but avoids the potential influence of the fluorescent reporter on the function of the intrabody due to genetic fusion. Transfections of these new constructs in cultured rat neonatal cardiomyocytes resulted in similar observations to those made for the non-fluorescently tagged intrabody constructs (Supplementary Fig. 5). A small difference in the Ca$^{2+}$ peak amplitude was, however, observed between the negative control VHH$_2$ Cas9-T2A-mCherry and untransfected rat neonatal cardiomyocytes,

underscoring the importance to include this non-relevant transfection control.

Freshly isolated adult cardiomyocytes were loaded with Fluo-4 AM calcium-binding dye and plated onto glass-bottom culture dishes coated with murine laminin. Detection of mCherry fluorescence allowed us to identify intrabody-expressing cells. Cells from the same isolation that were negative for mCherry expression were recorded and served as an internal control.

Calcium flux recordings were subsequently performed using electrical pacing at 1 Hz and 4 Hz (Fig. 5b-e). We found significant effects on the average relaxation rate ($1/\tau$) and the $T_{50}$ value of the Ca$^{2+}$ transients for cells expressing the intrabody VHH$_2$ B4B4-T2A-mCherry compared to untransfected cells or cells expressing the non-relevant intrabody control VHH$_2$ Cas9-T2A-mCherry (Fig. 5f, g, Supplementary Table 4). At 1 Hz

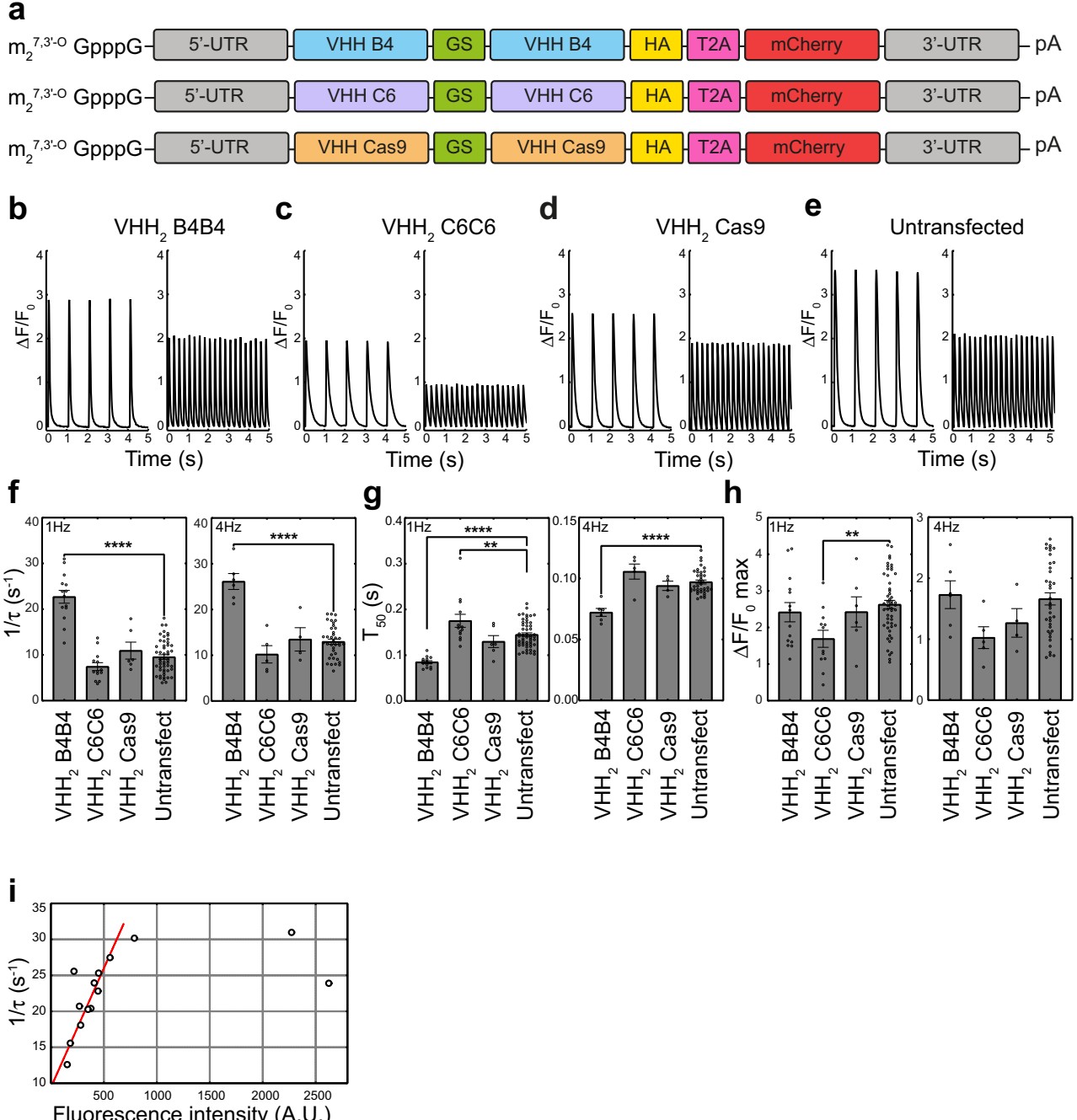

**Fig. 5 Effects of modRNA mediated intrabody expression on calcium cycling in adult cardiomyocytes. a** modRNA construct design. $m_2^{7,3'\text{-}O}$ GpppG is the ARCA CAP; 3'UTR is the 3' untranslated region, VHH is the intrabody encoding sequence, GS is a sequence that encodes a glycine-serine linker peptide (GGGSGGGSGGGSGGGS), HA is the sequence that encodes the hemagglutinin tag, T2A is the T2A peptide encoded sequence, mCherry is the mCherry sequence, 5' UTR is the 5' untranslated region and pA is the poly-A tail. **b–e** Representative $Ca^{2+}$ transients of each of the groups at 1 Hz and 4 Hz (VHH$_2$ B4B4: $n = 14$ at 1 Hz, $n = 6$ at 4 Hz; VHH$_2$ C6C6: $n = 13$ at 1 Hz, $n = 5$ at 4 Hz; VHH$_2$ Cas9: $n = 6$ at 1 Hz, $n = 4$ at 4 Hz; untransfected cells: $n = 52$ at 1 Hz, $n = 38$ at 4 Hz) **b** cells transfected with modRNA encoding for VHH$_2$ B4B4, **c** cells transfected with modRNA encoding for VHH$_2$ C6C6, **d** cells transfected with modRNA encoding for VHH$_2$ Cas9, **e** Untransfected cells. **f–h** Individual data points (open circles) and average value (bar graph) for the parameters for $1/\tau$ (rate of transient decay), $T_{50}$ (50% time elapse or ($Ca^{2+}$ transient width) and peak height $\Delta F/F_{0\ max}$ ($Ca^{2+}$ transient amplitude) (VHH$_2$ B4B4: $n = 14$ at 1 Hz, $n = 6$ at 4 Hz; VHH$_2$ C6C6: $n = 13$ at 1 Hz, $n = 5$ at 4 Hz; VHH$_2$ Cas9: $n = 6$ at 1 Hz, $n = 4$ at 4 Hz; untransfected cells: $n = 52$ at 1 Hz, $n = 38$ at 4 Hz). Statistical significance between groups was evaluated using one-way ANOVA and two-tailed Tukey-Kramer post-hoc tests assuming normal distribution. Error bars represent the standard error of the mean (SEM). Significance levels for differences of treated groups with the control group (untransfected cells) are indicated: **$P$-value < 0.01, ****$P$-value < 0.0001. **i** Scatter plot of the $1/\tau$ (rate of $Ca^{2+}$ transient decay) values for VHH$_2$ B4B4-T2A-mCherry transfected cardiomyocytes vs mCherry intensity. A linear regression trend line through the first 5 points is shown as a dotted line. A.U. arbitrary units. Source data are provided as a Source Data file.

electrical pacing, the average $T_{50}$ value decreased 1.5–1.7 fold for the VHH$_2$ B4B4-T2A-mCherry transfected cells compared to control cells (Fig. 5g, Supplementary Table 4) (Untransfected, $P < 0.0001$; VHH$_2$ Cas9-T2A-mCherry transfected cells, $P = 0.023$). Even more significantly, the average relaxation rate $1/\tau$ of the Ca$^{2+}$ transients of cells positive for VHH$_2$ B4B4 increased by 2.0–2.4 fold, compared to untransfected cells and cells transfected with VHH$_2$ Cas9-T2A-mCherry (Fig. 5f, Supplementary Table 4) ($P < 0.0001$). We observed similar significant differences at 4 Hz electrical pacing, but with overall higher values for the average $1/\tau$ and smaller values for the average $T_{50}$ times (Fig. 5f, g, Supplementary Table 4). We did not observe any significant difference between the average Ca$^{2+}$ transient peak height of the VHH$_2$ Cas9-T2A-mCherry transfected or untransfected cells to those that expressed VHH$_2$ B4B4-T2A-mCherry (Fig. 5h, Supplementary Table 4).

In contrast to VHH$_2$ B4B4-T2A-mCherry, the pS16 targeting intrabody, VHH$_2$ C6C6-T2A-mCherry, had an opposite but modest effect on the Ca$^{2+}$ transients of mature murine cardiomyocytes. Our analysis revealed a significant difference in the average Ca$^{2+}$ peak amplitude ($\Delta F/F_{0\ max}$) compared to that of the untransfected cells at 1 Hz pacing conditions ($P = 0.0058$) (Fig. 5h, Supplementary Table 4), as well as a moderate increase for the average $T_{50}$ value at 1 Hz electrical pacing compared to both control groups ($P = 0.014$). However, in difference to the cells expressing VHH$_2$ B4B4-T2A-mCherry, we did not find significant changes in any of the other measured parameters, although a consistent slight reduction in the average relaxation rate ($1/\tau$) compared to control cells (Fig. 5f, Supplementary Table 4) was noted in both 1 Hz and 4 Hz pacing conditions.

The transfection of the intrabody-T2A-mCherry modRNA constructs resulted in stoichiometric co-expression of mCherry. Therefore, the fluorescence intensity of mCherry is indicative of the intrabody expression levels. Relative expression levels of the VHH$_2$ B4B4 intrabody correlates with the observed increase in the individual relaxation rates $1/\tau$, which plateau at mCherry fluorescence values >300 AU (Fig. 5i), consistent with a causal relationship between the expression of VHH$_2$ B4B4 and the magnitude of the observed lusitropic effect.

Altogether, we found that the intrabodies were highly expressed in adult wild-type mouse hearts using direct injection of modRNA encoding for these intrabodies. We further observed that freshly isolated cells expressing the VHH$_2$ B4B4 intrabody showed a remarkably enhanced lusitropy compared to the untransfected cells or cells that had been transfected with a control intrabody. Mature cardiomyocytes expressing the pS16 phospho-specific VHH$_2$ C6C6 produced only moderate inhibitory effects on inotropy and lusitropy in the murine adult cardiomyocytes.

**In vivo delivery of AAV9 vectors encoding for VHH$_2$ B4B4 results in increased contractility in the MLP knock out (MLP KO) heart failure mouse model.** Functional studies in rat neonatal and adult murine cardiomyocytes using modRNA-encoded intrabodies identified the pentameric PLN targeting VHH$_2$ B4B4 intrabody as the best candidate for improved cardiac function, based on enhanced lusitropy and inotropy, in a disease model. To test its therapeutic potential, we employed the naturally cardiotropic AAV9 vector[42] coding for the intrabody VHH$_2$ B4B4 as a T2A linked genetic fusion with the gene encoding the fluorescent reporter ZsGreen and the Troponin-T promoter, TNT455, for heart-specific expression (Fig. 6a). An identical vector without the intrabody coding sequence was chosen as a control vector. We administered these vectors to MLP KO mice (MLP$^{-/-}$), an established murine heart failure model for dilated

cardiomyopathy (DCM)[43]. We chose to use AAV9 based vectors over a modRNA approach as the latter results in a transient protein production and also requires the more invasive direct cardiac administration. Furthermore, although the modRNA injection leads to high transient expression, transfection is mainly concentrated in the immediate vicinity of the injection site. Transduction of AAV9 vectors leads to lower and slower onset but more sustained expression[44]. In contrast to modRNA, the vector can be administered via the less invasive tail-vein injection enabling uniform transduction due to the distribution by the circulatory system. Additionally, the presence of a cardiac-specific promoter minimises the expression of the intrabody in other non-cardiac tissues, which may help reduce any potential confounding physiological effects.

$MLP^{-/-}$ mice were treated with a single intravenous injection of $10^{13}$ AAV/kg vector particles, and 14 days later hemodynamic parameters were measured before the animals were terminated (Fig. 6c-e). We chose a transduction period of 14 days, as we have previously shown that AAV9 transgene expression is maximal at this time[45,46], while minimising the risk of an adverse immune response against the non-endogenous protein VHH$_2$ B4B4-T2A-ZsGreen.

We compared the mice treated with the VHH$_2$ B4B4-T2A-ZsGreen AAV9 vector to those of the control group, treated with the AAV vector carrying only the reporter gene ZsGreen. The intrabody treated group showed significantly increased contractility (d$P$/d$T_{max}$) ($P = 0.037$) (Fig. 6c, Supplementary Table 5) and relaxation rate (d$P$/d$T_{min}$) (Fig. 6d, Supplementary Table 5) compared to those of the control group ($P = 0.0038$). The relaxation time ($\tau$), was also decreased in the group treated with the VHH$_2$ B4B4-T2A-ZsGreen AAV9 vector compared to the AAV9 ZsGreen control group ($P < 0.011$, Fig. 6e, Supplementary Table 5). Western blot analysis confirmed VHH$_2$ B4B4 expression, with some variability among different individual mice, whereas PLN levels remained unaffected (Fig. 6f). In addition, we also evaluated the expression levels of the transgene in liver tissue and quadriceps muscle taken from the same mice, showing no detectable expression of the VHH$_2$ B4B4 intrabody, emphasising the tight control of the TNT455 cardiac-specific promotor (Fig. 6g).

Taken together, we demonstrated that the delivery via a single intravenous injection of AAV9 vector encoding for VHH$_2$ B4B4 had a significant positive impact on the contractility and relaxation of the heart muscle in a heart failure model.

## Discussion
In this study, we have presented a broadly applicable and efficient strategy for intrabody discovery using in vitro selection of synthetic VHH phage-display libraries and rapid functional characterisation using in vitro and in vivo modRNA transfection assays. We selected the imperative heart failure target, PLN, a challenging target for small molecule approaches, as a case study for our strategy. With our robust pipeline, we identified a PLN-specific intrabody VHH$_2$ B4B4, which showed significant beneficial effects on cardiac contractility and relaxation in a murine heart failure model (Fig. 6), with a clear understanding of its mechanism of action.

Since PLN is highly conserved across mammalian species, we used a synthetic VHH library to obtain high-affinity PLN binders over the more commonly used immunisation approaches[27,47]. Moreover, the activity of PLN is regulated by a complex mechanism involving self-association into pentamers, by phosphorylation, and by allosteric mechanisms coupling PLN to the catalytic cycle of SERCA2a[4,17]. This provided an opportunity to showcase the power of synthetic VHH libraries to identify VHHs that target different functional states of the

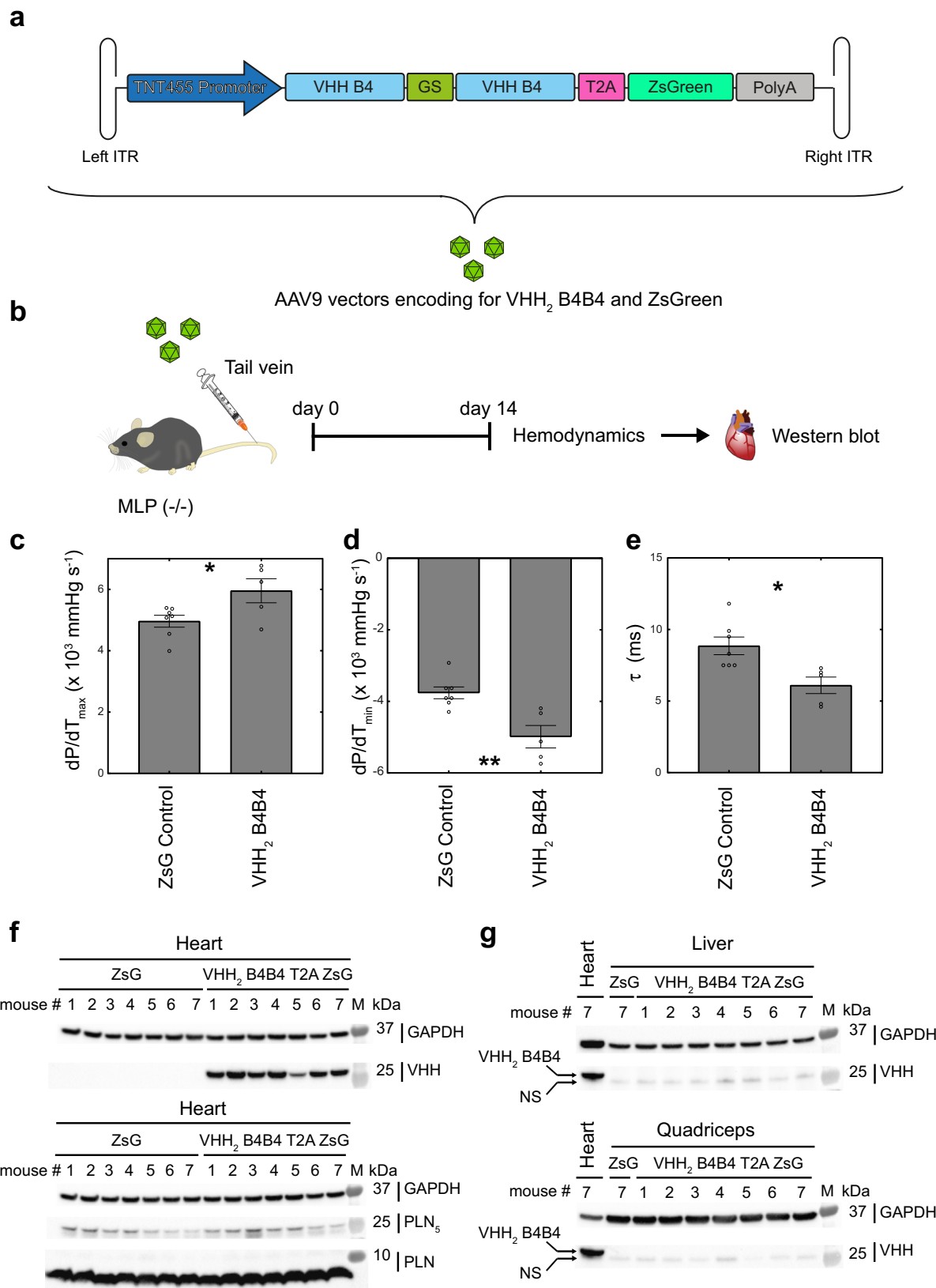

protein, including PLN pentamers and phosphorylated states of the protein. Our approach resulted in the successful isolation of PLN binders that can also distinguish between phosphorylated and non-phosphorylated forms of PLN, which was all achieved in vitro using peptide-based selections for binding. Furthermore, with the bivalent VHH$_2$ B4B4 variant, we were able to increase the apparent affinity of the monomeric intrabody by 300-fold for binding to non-phosphorylated PLN and by 600-fold for binding to pS16PLN. This relative ease by which manifold VHH constructs with enhanced properties can be generated has also been reported for VHHs with different specificities[48,49].

**Fig. 6 Hemodynamic analysis of VHH$_2$ B4B4-T2A-ZsGreen AAV9-treated mice versus control (ZsGreen AAV9). a** AAV9 vector design. Left ITR and Right ITR refer to the left and right inverted terminal repeat, TNT455 promoter is the Troponin-T heart-specific promoter, VHH B4 is the VHH B4 intrabody gene sequence, GS is a DNA sequence that encodes a glycine-serine linker peptide (GGGSGGGSGGGSGGGS), T2A is the T2A peptide encoded gene sequence, ZsGreen represents the ZsGreen reporter gene sequence. **b** Overview of the procedure: $MLP^{-/-}$ mice were injected with vehicle ZsGreen AAV9 ($n = 7$) or VHH$_2$ B4B4-T2A-ZsGreen AAV9 ($n = 5$) and hemodynamic parameters were assessed in acute terminal experiments 14 days later. **c–e** Individual data points (open circles) and average values (bar graph) of contractility (pressure increase velocity ($dP/dT_{max}$), relaxation (pressure decrease velocity $dP/dT_{min}$), and relaxation time ($\tau$). Significant differences are indicated and evaluated using two-tailed student $t$-test. Error bars represent the standard error of the mean (SEM). Significance level between groups are indicated: *$P$-value < 0.05, **$P$-value < 0.01. (exact $P$-values in Supplementary Table 5). **f** Western blot analysis of heart samples of AAV9 ZsGreen control mice (ZsG) ($n = 7$) and the AAV9 VHH$_2$ B4B4-T2A-ZsGreen (VHH$_2$ B4B4-T2A-ZsG) group ($n = 7$). Membranes were stained with anti-VHH and anti-GAPDH, (Top) and with anti-PLN and anti-GAPDH (bottom). Pentameric PLN is indicated as PLN$_5$. **g** Liver (top) and quadriceps (bottom) tissue samples from mice of the VHH$_2$ B4B4-T2A-ZsGreen group (VHH$_2$ B4B4-T2A-ZsG). A liver (top) or quadriceps (bottom) sample of mouse #7 of the control group (ZsG) was added as negative control and a heart sample from mouse #7 of the AAV9 VHH$_2$ B4B4-T2A-ZsGreen treated group was used as a positive control (indicated above the lane). Arrows labelled "VHH$_2$ B4B4" and "NS", mark the bands for VHH$_2$ B4B4 and a faint non-specific band present in all samples including the negative control. Sample identities are indicated above the lanes for all blots. The sizes of the molecular weight marker "M" are noted on the right side of the membrane with the primary antibody used to stain the membrane. Source data are provided as a Source Data file.

In agreement with other studies[29], we confirmed that all of our PLN binding VHHs, in both monomeric and dimeric form, express very well inside the cell and engage functionally with their target in the cytosol, whilst demonstrating no noticeable associated toxicity; indicating that they are stably folded and well-tolerated with minimal off-target effects. Immunoprecipitation studies further confirmed that the avidity effect of the bivalent VHH$_2$ B4B4 results in a strong binding preference for the native pentameric state of PLN within the cell, proving that VHH dimerisation is a successful approach for targeting homooligomeric states of its target protein. Taken together, we have demonstrated that by using synthetic VHH libraries, we can isolate potent PLN-specific intrabodies that have excellent expression profiles in cells and tissue. Furthermore, we have shown that synthetic VHHs can be selected to successfully target different functional states of the highly conserved PLN protein.

In our study, we were also able to demonstrate the tremendous benefit of utilising modRNA technology for both the rapid biochemical and phenotypic validation of our intrabody candidates. RNA-based transfection methods have until recently been held back in biomedical research due to the high immunogenicity of mRNA. Modified mRNAs, whereby bases have been chemically modified to escape the recognition by Toll-like receptors, have been shown to have reduced immunogenicity and improved stability[38,39]. ModRNA has since been demonstrated to be valuable for immunisation strategies[50], for secretion and display of antibodies[51,52] and for the secretion of paracrine factors[37,53]. The utility of intrabody modRNA transfection for cell imaging purposes has also been shown[54].

As a nucleotide-based drug modality, modRNA could provide an alternative strategy to DNA based methods and bypassing the potential safety risk of possible DNA genome integration. However, modRNA transfection typically results in very transient protein expression. This pulse-like transfection/expression kinetics of modRNA, is a significant advantage for the expression of secreted proteins, especially paracrine factors which are required to be switched on and off rapidly[37,55]. In contrast, this transient nature of modRNA expression may significantly limit their use in some instances for intrabodies, as some targets and tissues may require sustained and uniform intrabody expression to show benefit. Our study demonstrated the use of the less invasive and more homogenous transduction via a naturally cardiotropic AAV9 vector under the control of a strong heart-specific promotor for intrabody expression. This resulted in a prolonged-expression of the intrabodies specifically in the heart. This tissue specificity minimises the risk of adverse effects arising from cross-reactivity with other targets. This issue might be of particular relevance for intrabodies that recognise linear epitopes, such as our anti-PLN intrabodies, as these epitopes are likely to contain sequence motifs that are present in other proteins. Therefore, the AAV delivery proved to be a better option to assess the physiological effects of our PLN-specific intrabodies compared to the approach of intracardiac injection of ModRNA. Increasing technological progress in formulations and instrumentation, however, i.e. Lipid Nanoparticles (LNPs)[56] and novel generations of RNA modifications to increase lifetimes and translatability[57], are under development to address these limitations along with catheters and implantable devices for less invasive local and sustained delivery[58].

Taken together, we have shown that screening for expression, target engagement and phenotypic effects in relevant cell and tissue models using small ModRNA libraries allows efficient intrabody candidate triaging for the progression of the best candidate towards the more elaborate process of AAV9 vector development. This in turn led to the clear observation of the improved heart function at the physiological level in a mouse heart failure model due to the expression of the PLN inhibiting intrabody.

By generating intrabodies with specificities for different states of PLN, we have developed precision tools that can elucidate how various forms of PLN contribute to the regulation of calcium flux within the heart. The intrabodies VHH B4 and VHH C6 differ by targeting either a pan PLN state or the most abundant phosphorylated state, pS16 PLN. By engineering the intrabodies VHH B4 and VHH C6 into the bivalent variants VHH$_2$ B4B4 and VHH$_2$ C6C6, we were able to generate molecules that have strongly enhanced binding potencies and preferentially bind to pentameric conformers of PLN. In Fig. 7, we outline the proposed mechanism of action of the different intrabodies.

In the absence of the intrabodies, PLN exists in a dynamic equilibrium between a pentameric and a monomeric state capable of directly binding to SERCA2a and inhibiting Ca$^{2+}$ ATPase activity by lowering the affinity of SERCA2a for Ca$^{2+}$ ions. In its phosphorylated state, PLN no longer inhibits the activity of SERCA2a. This occurs through an allosteric mechanism involving the interaction of the regulatory domain of PLN with an allosteric site on SERCA2a[17,18] (Fig. 7a).

Expression of the intrabody VHH B4 or VHH$_2$ B4B4 exhibited a significant lusitropic effect in rat neonatal and adult mouse cardiomyocytes. We were able to observe a clear correlation between VHH$_2$ B4B4 expression levels and the magnitude of the induced lusitropic effect in wild-type adult murine cardiomyocytes (Fig. 5i). Surprisingly, the enhanced SERCA2a activity in

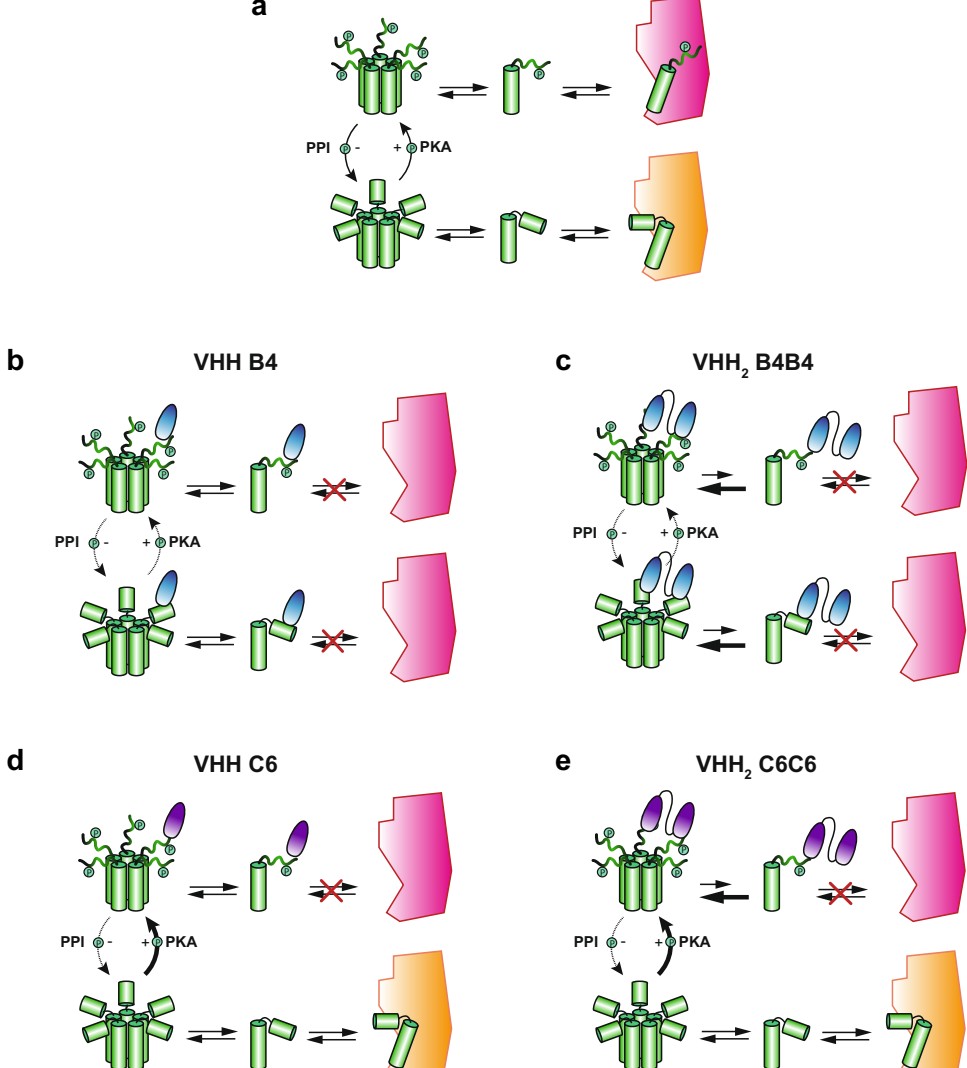

**Fig. 7 Molecular mechanism of action of the different PLN-specific intrabodies. a** Molecular mechanism of PLN function. Pentameric PLN (green) is in dynamic equilibrium with PLN monomers, which bind and inhibit SERCA2a (inhibited SERCA2a, yellow; activated SERCA2a, red). PKA phosphorylates PLN and PPI dephosphorylates PLN. Phosphorylated PLN monomers are able to bind to active SERCA2a. **b** VHH B4 and **c** $VHH_2$ B4B4 bind to all forms of PLN, but with differing affinities. This binding results in steric hinderance of PLN or pS16 PLN to bind to SERCA2a, leading to activation of SERCA2a. $VHH_2$ B4B4 additionally shifts the equilibrium toward pentameric species of PLN (indicated by a larger arrow pointing towards the shift in the equilibrium). Steric hinderance may lead to inhibition of PLN phosphorylation by PKA, the binding of the pS16 PLN species by VHH B4 or $VHH_2$ B4B4 may also protect pS16 PLN from dephosphorylation by PPI. **d** The VHH C6 and **e** $VHH_2$ C6C6 bind only to phosphorylated PLN and phosphorylated $PLN_5$, leaving the binding equilibria of the unphosphorylated PLN state unaltered. Steric hinderance inhibits the association of pS16 PLN to SERCA2a, the free SERCA2a can bind to unphosphorylated PLN leading to SERCA2a inactivation. This is however offset by protection of PLN dephosphorylation by VHH C6 or $VHH_2$ C6C6 from PPI activity and accumulation of pS16 PLN. A dotted arrow for the PPI activity and a thicker arrow for the kinase reaction indicate this perturbation of the PLN phosphorylation balance. $VHH_2$ C6C6 additionally shifts the equilibrium towards pS16 $PLN_5$ due to the stabilisation of the pentameric form (indicated by a larger arrow pointing towards the shift in the equilibrium).

VHH B4 or $VHH_2$ B4B4 expressing cells (in both rat neonatal cardiomyocytes and adult mouse cardiomyocytes) did not result in increased amplitude of the $Ca^{2+}$ transient, suggesting that SR $Ca^{2+}$ content is normalised in these cells compared to the untransfected cells. This effect might originate from a more sensitised ryanodine receptor leading to an increased SR leak or spark frequency. This mechanism is consistent with an increased $Ca^{2+}$ release rate for rat neonatal cardiomyocytes transfected with VHH B4 and $VHH_2$ B4B4. Interestingly, our observations are consistent with the previously reported scFv intrabody, which was derived from immunised chickens, suggesting that this is a general mechanism for specific targeting of PLN with intrabodies[59,60].

Although VHH B4 can bind to all forms of PLN, inhibition of PLN function is most likely only achieved through binding to monomeric PLN (Fig. 7b). By contrast, $VHH_2$ B4B4 has the combined effect of stabilising the PLN pentameric species as well as directly affecting the interaction of monomeric PLN molecules to bind to SERCA (Fig. 7c). This increases the fraction of free SERCA2a molecules and leads to significantly enhanced $Ca^{2+}$ uptake, consistent with the observation that $VHH_2$ B4B4 had superior inhibitory effects on PLN function compared to the other VHHs. Additionally, albeit with 10 fold lower affinity, $VHH_2$ B4B4 and VHH B4, also bind to the pS16 phosphorylated state of PLN, a species known to induce both lusitropy and inotropy by increasing SERCA activity[4]. Binding of the VHHs to

PLN pS16 could protect it from dephosphorylation by PPI, mimicking the effects and proposed mechanism of multivalent binding of 14-3-3 proteins in the heart[61].

The expression of the phospho-specific intrabodies VHH C6 and VHH$_2$ C6C6 resulted in only minor effects on cardiomyocyte calcium kinetics, e.g. the decay phase, leading to a more speculative model for the mode of action of these intrabodies. A number of studies have demonstrated that the relief of SERCA2a inhibition by phosphorylation of PLN proceeds through an allosteric mechanism whereby fully bound PLN forms additional interactions with a regulatory site on SERCA located in the adaptor domain[17,62]. Targeting these interactions could potentially lead to renewed inhibition of SERCA2a (Fig. 7c, d). Complete dissociation of pS16 PLN, however, will lead to activation of SERCA2a. Therefore, even though VHH C6 and VHH$_2$ C6C6 bind to the phosphorylated state of PLN specifically, the lack of significant inhibition is consistent with the dissociation of phosphorylated PLN monomers from the pS16PLN:SERCA2a complexes. As the intrabodies VHH C6 and VHH$_2$ C6C6 do not bind to non-phosphorylated PLN, the non-phosphorylated PLN molecules will be able to occupy the resulting free SERCA2a molecules, leading to a small increase of inhibited SERCA2a depending on the size of the increased free SERCA2a population. The potential protective action of VHH$_2$ C6C6 and VHH C6 will, however, lead to the protection of pS16 PLN, again offsetting this effect (Fig. 7c, d). An in-depth analysis of the Ca$^{2+}$ dynamics in cardiomyocytes transfected with VHH C6 and VHH$_2$ C6C6, using catecholamines to modulate the phosphorylation state of PLN, will be required to provide further validation of this model.

The biology for the various states of PLN is complex, affecting key pathways including degradation and transcriptional activation[63]. The intrabodies developed in this study will therefore be valuable tools in future research to dissect the functional outcomes of targeting PLN. From a future therapeutic point of view, the selectivity that intrabodies offer can potentially enable the targeting of complex disease-specific regulatory mechanisms while mitigating any negative consequences that result from the complete removal of a protein, offering a more buffered approach to modulating the regulome of this complex protein network. As such, when combined with emerging viral or nucleic acid delivery technologies, VHH intrabodies present a unique opportunity to expand the pharmacological toolbox beyond currently applied genetic approaches.

## Methods

**Animal experiments**. Animal studies at Karolinska Institutet were carried out in accordance with the institutional guidelines, and all animal experiments were approved by the local ethics committee (Stockholm, Sweden) in accordance with the Animal Protection Law, the Animal Protection Regulation, and the Regulation of the Swedish National Board for laboratory animals. Animal studies at AstraZeneca were performed in accordance with the National Institute of Health (NIH) guidelines for use of experimental animals and the study protocol was approved by the Animal Ethics Committee at Gothenburg University (Gothenburg Ethical Review Board number Ea001173-2017). The animals were housed in a temperature-controlled (20–23 °C) facility with a 12 h light/dark cycle, relative humidity of 40–60% with 20 air changes/hour, in an American Association for the Accreditation of Laboratory Animal Care (AALAC)-approved facility. The animals had free access to pelleted food (R70, Lantmännen, Sweden) and water. Cages contained hardwood bedding (J.Rettenmeier & Söhne GMBH, Germany), shredded paper (Papyrus AB, Sweden), a plastic house and gnawing sticks (Tapvei, Estonia).

**Peptide design**. We designed a series of peptides (Bachem, Switzerland) based on the amino-acid sequence of the cytoplasmic domain of PLN (Table 1). These peptides were used in panning experiments of VHH synthetic libraries as well as for the subsequent screening and biophysical characterisations.

**Preparation of peptide coated streptavidin dynabeads**. A solution containing 4 mM N - [6 - (Biotinamido) hexyl] - 3´- (2´-pyridyldithio) propionamide (EZ link HPDP-Biotin) (Thermo Fisher Scientific, USA) was prepared in DMSO and 50 µl

| Table 2 Amino-acid sequences of the PLN-specific VHHs, VHH B4 and VHH C6. | |
|---|---|
| **Name** | **Sequence**[a] |
| VHH B4 | EVQLQESGGGLVQAGGSLRLSCAASGFSFYRHAMGWFRQAPGKE REWVAEIDWEGGYTYYADSVKGRFTISRDNAKNTVYLQMNSLKP EDTAVYYCAAGRDLYAYWGQGTQVTVSS |
| VHH C6 | EVQLQESGGGLVQAGGSLRLSCAASGFTLYAMGWFRQAPGKERE FVAAITPHGSTTYYADSVKGRFTISRDNAKNTVYLQMNSLKPED TAVYYCHTYYSGSWGQGTQVTVSS |

[a]VHH amino-acid sequences are shown in single letter code.

reagent was added to 1 mg of M270 streptavidin dynabeads (Thermo Fisher Scientific, USA) in 1 ml of phosphate-buffered saline (PBS) buffer and incubated (rotating) overnight to allow cysteine reactive groups to be displayed on the beads. After washing the beads three times with PBS and using a strong magnet, the unabsorbed biotin-HPDP reagent was removed. Subsequently, the beads were incubated with 10 µM of either peptide I or III (Table 1) and incubated (rotating) overnight at room temperature to allow the reaction of the cysteine residues with the HPDP moiety. Then the beads were washed finally thrice with PBS, ready for use in panning experiments.

**Isolation of VHH binding to PLN derived peptides from synthetic VHH phage-display libraries**. The PLN-specific VHHs B4 and VHH C6 were isolated from a LlamdA® phage-display VHH library[64,65], built using Colibra® technology, which was licenced from the company Isogenica (Saffron Walden, UK). This library is made up of five sub-libraries, built on a llama VHH framework with fixed lengths of CDR1 and CDR2, but which differ in CDR3 lengths ranging from 5 to 21 residues[64].

Sub-library 1 was subjected to three rounds of panning using the streptavidin dynabeads with cysteine-linked peptides III and I. At each round M270 streptavidin dynabeads (Thermo Fisher Scientific, USA) and 10$^{12}$ input phages were blocked for 1 h rotating with 5% skimmed milk in Tris Buffered Saline, 0.1% Tween 20 (TBST) and the blocked phage was then incubated with 100 mg of blocked streptavidin beads for 1 h rotating at room temperature. The beads were then separated using a magnetic rack (DynaMag™-2, Thermo-Fisher, UK) and the supernatant phage was incubated with 100 mg of cysteine-linked PLN peptide III, or I in the first round of selection or with blocked streptavidin beads (negative control). After 1 h incubation at room temperature, the beads were washed six times with PBS and six times with PBST, using the magnetic rack to separate beads and liquid. Subsequently, elution of the phage was accomplished by a 10 min incubation of the washed beads with 10 mM dithiothreithol (DTT) in PBS at room temperature and by separation of beads and phage using the magnetic rack. Output phage from the peptide selection and the negative control were titrated to determine the phage-titre and the enrichment factor. The remaining phage output from the positive selection was used to infect TG1 E. coli cells, which, after plating and overnight growth, were harvested in media supplemented with ampicillin and 50% glycerol. The phage was subsequently rescued and used as input for the next rounds of panning.

Rounds 2 and 3 were performed in an identical way, with the exception that the amount of PLN-streptavidin beads was halved after each round. After the third round of panning, half of the eluted phage was used to infect HB2151 E. coli cells and individual colonies were picked and grown in Terrific Broth medium supplemented with 0.1% Glucose and 100 µg/ml ampicillin under continuous agitation at 37 °C in 1 ml 96 deep well plates. After 4 h of incubation, cultures were induced with 1 mM IPTG overnight at 30 °C to allow for the expression of the VHHs for binding detection using ELISA. Both outputs from the panning using peptides I and III were evaluated for binding to peptide IV, which lead to the identification of clone B4. ELISA experiments using peptide VI captured on Nunc maxisorb 96-well ELISA plates (Thermo Fisher, USA) coated with 2 µg/ml of neutravidin in PBS (overnight, RT), resulted in the identification of clone C6. The sequences of these two VHHs are given in Table 2.

**Construction and expression of the bivalent VHH$_2$ B4B4**. For the generation of the bivalent VHH$_2$ B4B4, two PCR reactions were performed. For the amplification of the N-terminal domain we constructed a pHEN4 specific forward primer 5'-CCC-AGG-CTT-TAC-ACT-TTA-TGC-TTC-3' that anneals in the vector upstream of the VHH gene, and a reverse primer containing a BamHI restriction enzyme site (underlined) 5'-TTT-GGA-TCC-TCC-GCC-GCT-GCT-CAC-CGT-AAC-CTG-GGT-3', encoding for seven residues of framework 4 of the VHH domain and the first three residues of a (Gly$_3$Ser)$_3$ linker domain. The resulting PCR product was digested with NcoI and BamHI and purified using Qiagen PCR purification kit (Qiagen, UK). A second PCR reaction was performed using a forward primer 5'-TCT-TGG-ATC-CGG-CGG-AGG-TAG-TGG-CGG-AGG-TAG-TGA-AGT-TCA-GCT-GCA-AGA-AAG-CGG-3', containing a BamHI site (underlined) and encoding for the rest of the (Gly$_3$Ser)$_3$ linker and the first eight

residues of framework 1, and a VHH reverse primer 5'-CAC-AAC-GCC-TGT-AGC-ATT-CCA-C-3', annealing in the pHEN4 vector downstream of the VHH gene. The resulting PCR product was digested with *BamHI* and *NotI* and purified using Qiagen PCR purification kit (Qiagen, UK). The pHEN4 vector was digested with *NotI* and *NcoI* and purified. The bivalent construct was obtained by a three-point ligation reaction of the two purified PCR fragments and the digested pHEN4 vector and used to transform HB2151 *E. coli* cells. Individual colonies were picked, and purified plasmids were sequenced.

**Surface plasmon resonance measurements.** Clones VHH B4, VHH₂ B4B4 and VHH C6 were expressed and purified according to previously described protocols[66]. VHH concentrations were determined by using UV absorption spectrophotometry and using sequence-based calculated extinction coefficients[67] at 280 nm. Kinetic rate constants, $k_a$ and $k_d$ and the dissociation constant $K_d$ for the VHH PLN interactions were measured using a Biacore T200 instrument (Cytiva, UK). Streptavidin chips SA (Cytiva, UK) were used to immobilise 50–100 pg/mm² (50–100 resonance units) of either peptide V or VI (Table 1) in flow cell 2 of the chip. Flow cell 1 was left unaltered and served as a reference cell. Binding kinetics was either measured using single-cycle kinetics using five increasing concentrations and a zero concentration in PBS buffer, pH 7.4 + 0.05% Tween 20. Association traces were recorded for 3 min and dissociation of the complexes was followed for 5–10 min. Curves obtained after subtraction of the reference and buffer signals were fitted to a 1:1 Langmuir binding model with the program biaeval 3.1 (Cytiva, UK) for the PLN binding reactions of the monomeric VHHs, VHH B4 and VHH C6, and to a bivalent analyte model for the VHH₂ B4B4:PLN interaction.

**ModRNA library design and synthesis.** Modified mRNA of monomeric or bivalent VHHs, including a range of C-terminal tags: hemagglutinin (HA) tag, HA-mCherry, HA-mCherry-PEST or T2A-mCherry, were synthesised in vitro using T7 RNA polymerase-mediated transcription from a linearised synthetic DNA template, which incorporates generic 5' and 3' UTRs and a poly-A tail, as previously described[68,69] using Megascript T7 transcription kit (Thermo Fisher Scientific, SE). For all these RNAs, uridine was fully replaced by N1-methylpseudouridine (Trilink, USA) in the in vitro transcription reaction. The RNA was purified using Ambion MEGA clear spin columns (Thermo Fisher Scientific, SE) and treated with Antarctic Phosphatase (New England Biolabs, SE) for 30 min at 37 °C to remove residual 5'-phosphates. The RNA was then re-purified and quantified by Nanodrop (Thermo Fisher Scientific, SE). After purification, modRNA was resuspended in 10 mM Tris HCl, 1 mM EDTA at 1 µg/µl for use in in vitro experiments. For the use of in vivo experiments, large-scale batches of modRNA (400 µg) were concentrated to 10 µg/µl using ethanol precipitation.

**Mammalian cell culture and cell transfection.** HeLa cell lines were cultured in DMEM media (Thermo Scientific, UK) supplemented with 10% FCS (Thermo Scientific, UK) and 5% Glutamax (Thermo Scientific, UK), and plated and propagated according to standard cell culture techniques. ES03 hESC (purchased from WiCell, USA) cells were cultured in E8 basal media (Thermo Scientific) and passaged every 3–4 days and plated on tissue culture plates coated with matrigel (Thermo Scientific). Human iPS derived cardiomyocytes (iPS CM) were purchased from FUJIFILM Cellular Dynamics, Inc. (FCDI) (cat. No. CMC-100-012-001), and frozen cells were thawed and plated according to the manufacturers' recommendations. Before transfection experiments, HeLa or hESC cells were plated at >70% confluence in 96- or 6-well culture plates. HeLa, hESC and iPS CM were transfected using RNAiMAX (Thermo Fisher Scientific, USA) mixed with various amounts of modRNA-encoded VHHs alone or in combination with modRNA constructs of phospholamban. ModRNA:RNAiMAX mixtures were made in a 1:1 w/v ratio in Optimem (Thermo Fisher Scientific, USA) and incubated for 4 h in a humidified chamber at 37 °C and 5% CO₂. Subsequently, the Optimem media was exchanged for the appropriate cell media and incubated for another 20 h in the same chamber.

**Incucyte measurements.** Plated and transfected cells with modRNA constructs containing a mCherry coding sequence were incubated in an Incucyte Zoom instrument (Essen Bioscience, USA). Live fluorescence imaging was performed using a ×10 objective and using LED fluorescence excitation @585 nm and an 635 nm emission filter. Fluorescence images were taken from a region of interest of wells of 96-well plates every 2 h of incubation.

**Cell viability assays.** Cell viability of iPS CM transfected with modRNA encoding for VHH B4-HA, VHH C6-HA or VHH Cas9-HA (non-relevant control) was determined using the CellTiter-Glo® Luminescent Cell Viability Assay and compared to control cells, treated only with the transfection reagents. Cell viability of iPS CM transfected with modRNA encoding for VHH₂ B4B4-T2A-mCherry, VHH₂ C6C6-T2A-mCherry, VHH₂ Cas9-T2A-mCherry, VHH₂ B4B4-HA, VHH₂ C6C6-HA was determined with a CellTiter-Glo® 2.0 Luminescent Cell Viability Assay (Promega) and compared to control cells that were only treated with the transfection reagents. The assays were performed following the manufacturer's protocols. Bioluminescence was measured using Envision plate reader with Ultrasensitive module (Perkin-Elmer) using 384-well plates.

**Isolation and cell culture of rat neonatal cardiomyocytes.** Six-days-old Sprague-Dawley rat pups (KI Huddinge in house colony) were sacrificed, and the hearts were collected in ice-cold calcium and magnesium-free PBS buffer. The atria and vessels were removed using a scalpel and the remaining ventricles were cut to pieces of ~1 mm. The cardiomyocytes were isolated using a MACS gentle separator (Miltenyi biotech, Germany) according to the manufacturer's recommendations. Cells resuspended in DMEM:M199 (4:1 v/v) + Y-27632 (Sigma–Aldrich, Germany) (5 µM) and cytosine β-d-arabino-furanoside (Sigma–Aldrich, Germany) (10 nM) medium (DMEM and M199, Thermo Scientific, USA) in six-well plates at 1.8 × 10⁶ cells per well. Cell media was changed to DMEM:M199 (4:1 v/v) medium after 2 days.

**ModRNA transfection of rat neonatal cardiomyocytes.** Freshly cultured cells were transfected with modRNA using RNAiMAX (Thermo Fisher Scientific, USA) or Viromer red transfection agent (Lipocalyx, Germany). ModRNA:RNAiMAX mixtures were made in a 1:1 w/v ratio in Optimem (Thermo Fisher Scientific, USA) following manufacturers' recommended procedures. Viromer Red transfection mixtures were made according to the manufacturer's procedures and using 0.4:1 w/v (Viromer: modRNA) ratio. Subsequently the transfection mixtures were added to the cell culture medium and incubated for 4 h in a humidified chamber at 37 °C and 5% CO₂. Afterwards, the media was changed to DMEM supplemented with 5% fetal calf serum (FCS) and 1% PenStrep, and then incubated for another 20 h in the same chamber.

**Cell lysis of cultured rat neonatal cardiomyocytes.** Rat neonatal cardiomyocytes were lysed using RIPA buffer (Thermo Fisher Scientific, USA) containing protease + phosphatase inhibitor cocktail (Pierce Thermo Fisher Scientific, USA). The protein concentrations in the lysates were quantified using a Bicinchoninic acid assay (BCA kit, Thermo Fisher Scientific, USA).

**Western blot.** For western blot analysis, samples containing 30 µg of total protein from cell or tissue lysates were mixed with loading sample (LDS) buffer (Thermo Fisher Scientific, UK) and either not boiled in non-reducing conditions; or boiled in the presence of a reducing agent (Thermo Fisher Scientific, UK). All samples were subjected to SDS-PAGE using 4–12% Bis-Tris Nupage gels (Thermo Fisher Scientific, UK) and then transferred to nitrocellulose membranes using semi-dry electrophoretic transfer (Biorad, USA) or using a dry transfer iBlot 2.0 system (Thermo Fisher Scientific, UK). Membranes were blocked for 1 h at room temperature in 5% Skimmed milk in TBST (Sigma, Germany) and subsequently incubated overnight at 4 °C with primary antibodies: HA detection: 1:1000 dilution of the rabbit anti-HA antibody C29F4 (Cell Signalling Technology, USA) in TBST + 5% BSA; PLN detection: 1:3000 of the mouse anti-PLN 2D12 (Thermo Fisher Scientific, USA) in TBST + 5% Skimmed milk; VHH detection: 1:500 of the MonoRab™ Rabbit Anti-Camelid VHH Antibody (Genscript, The Netherlands) in TBST + 5% Skimmed milk; β-tubulin detection: 1:1000 of the anti-β-tubulin antibody (MA5-16308-HRP, Thermo Scientific, USA); GAPDH detection: 1:5000 rabbit anti-GAPDH 14C10 (Cell Signalling Technology, USA) in TBST + 5% Skimmed milk; cofilin detection: 1:1000 of the Cofilin (D3F9) XP® Rabbit mAb (HRP Conjugate) (Cell Signalling Technology, USA) in TBS + 0.1 % Tween 20 + 5% BSA. Membranes were subsequently washed 5 × 5 min or 3 × 10 min with TBST and then incubated for 2 h at room temperature with 1:2000 dilution of the secondary antibody (anti-rabbit HRP, Cell Signalling Technology, USA) for detection of rabbit primary antibodies; anti-mouse HRP (Cell Signalling Technology, USA) for detection of mouse primary antibodies. Membranes were subsequently washed 5 × 5 min with TBST and then developed with ECL Pierce™ ECL Western Blotting Substrate (thermo Scientific, UK), imaged on a Biorad Gel doc imager (Biorad, UK). When SuperSignal West Pico PLUS Chemiluminescent Substrate (Thermo Scientific, UK) was used for detection, secondary antibodies were used at 1:50000.

**Immunoprecipitation studies.** Cell lysate samples containing 350 µg of total protein were incubated with 100 mg of streptavidin dynabeads (M270, Thermo Fisher Scientific, USA) or protein-G dynabeads (Thermo Fisher Scientific, USA) for 1 h (rotating) at room temperature. The supernatant was subsequently mixed with 5 µg of biotinylated Fab 3F10 (Sigma, Germany), overnight 4 °C rotating, for immunoprecipitation studies with streptavidin dynabeads, or 3.5 µg of the anti-HA antibody C29F4 (Cell Signalling Technology, USA), respectively. The following day, 100 mg of streptavidin dynabeads (For Fab 3F10 capture) or protein-G dynabeads (for C29F4 capture) was added and incubated for 1 h rotating. Beads were subsequently washed three times using an equal volume of RIPA buffer and beads were separated from the supernatant using a magnetic rack (DynaMag™-2, Thermo-Fisher, USA); each washing step was performed using a fresh eppendorf in order to avoid any carry over from the original sample. For simple detection of the captured proteins was the beads were boiled in 1 × LDS buffer (Thermo Fisher Scientific, USA) and 1 × NuPAGE reducing agent (Thermo Fisher Scientific, USA) for 5 min and the supernatant was collected by separating the beads using the magnetic rack. For the evaluation of the native state PLN, elution was achieved by incubation using 1 × LDS buffer without reducing agent, 10 min rotation at room temperature and bead separation using a magnet for streptavidin dynabeads. For

protein-G dynabeads, captured proteins were eluted by incubating the beads in 50 μl of 0.1 M Glycine (pH 3) for 10–15 min at room temperature and neutralised by adding 10 μl of Tris (pH 7.5). Eluted samples were then analysed by western blot.

**Calcium imaging of rat neonatal cardiomyocytes and analysis.** Rat neonatal cardiomyocytes were obtained as described above. For calcium imaging experiments, isolated cells were plated at 150,000–200,000 cells /cm$^2$ on no 1.5 Matek dishes with clear bottom coverslips (Thermo Fisher Scientific, UK). Using Viromer red transfection agent (Lipocalyx, Germany), cells were transfected using intrabody modRNA constructs.

After 24 h incubation, cells were washed with DMEM:M199 (4:1 v/v) media and incubated with 10 μM of Fluo-4 AM (Thermo Fisher, USA) for 30 min at 37 °C and 5% CO$_2$. Cells were subsequently washed twice using Hepes buffered Tyrodes solution (140 mM NaCl, 5.4 mM KCl, 5 mM HEPES, 5.5 mM Glucose, 0.5 mM MgCl$_2$, 0.4 mM NaH$_2$PO$_4$, 1 mM CaCl$_2$, pH 7.4) at 37 °C. Electrically induced (10 ms pulse length, 8–12 V, 1–4 Hz) calcium transient measurements using an Ionoptix cell pacer and plate electrodes (Ionoptix, The Netherlands) were subsequently recorded for 15–20 s and for 6–10 fields of view using a ×20 objective on a Nikon Eclipse wide-field microscope equipped with an CCD camera for high-speed acquisition (Andor Zyla 4.2 +, Oxford Instruments, UK). The image stack was analysed using the Nikon elements analysis software or by the open-source imaging software Fiji (ImageJ)[70]. Each cell was manually picked and a region of interest (ROI) was contoured using the auto-detection tool, manual inspection and the fluorescence intensities were measured.

Fluo-4 fluorescence intensities were plotted as $(F − F_0)/F_0$ ($\Delta F/F_0$), where $F$ is the fluorescence intensity and $F_0$ is the fluorescence intensity at the diastole. Peak analysis was performed using an in house written peak analysis programme (P.A.P.), to obtain the average number of peaks, average peak maxima and minima, $T_{50}$ and $T_{75}$, the upstroke ($dF/dt_{up}$) velocities. The $\tau$ values for each cell were calculated by fitting an exponential decay function to the decay phase of the averaged calcium transients using the software WinEDR (v.3.8.9).

**Immunofluorescence imaging of rat neonatal cells.** Cells were fixed with 4% PFA in PBS, permeabilised with ice-cold acetone and subsequently blocked in 1.1% BSA in PBS overnight. Cells were subsequently stained using anti-HA (C29F4, Cell Signalling Technology, USA) and anti-PLN (2D12, Thermo Fisher, USA) antibodies at a 1:400 dilution in blocking buffer, overnight. Cells were washed with blocking buffer and incubated with fluorescently labelled secondary antibodies and nuclei were stained with Gold Anti-fade Reagent with DAPI (Invitrogen, USA). Imaging was performed using a Zeiss axio observer fluorescence microscope (Zeiss, Germany). The use of the anti-PLN antibody 2D12 for immunofluorescence studies was validated using adult murine heart tissue (Supplementary Fig. 6).

**ModRNA transfection of mouse myocardium in vivo.** 9-12weeks old C57BL/6 mice (Taconic, Denmark) were injected intracardially with 50 μg modRNA formulated in RNAimax + optimem (Thermo Fisher Scientific) (2:1:1 w/v/v ModRNA:RNAimax:Optimem) for time-resolved expression profiling and distribution in cardiac tissue. The surgical procedure for intracardial injection was carried out as described previously[37]. For each time point (24 h and 72 h), $n = 6$ mice treated with modRNA and $n = 3$ mice treated with vehicle only. At the 24 h and the 72 h time points, the mice were sacrificed and hearts were excised and washed briefly in PBS. The hearts were then either snap-frozen in liquid nitrogen (for the purpose of western blot analysis) or incubated for 4 h in PBS + 4% PFA to fix the tissue in preparation for immunofluorescence staining.

**Immunofluorescence studies on transfected mouse hearts.** For immunofluorescence, fixed hearts were washed three times for 30 min in PBS and then fixed with 4% PFA for 2 h before submerging in 30% sucrose. Hearts were subsequently embedded in OCT and frozen on dry ice. The frozen block was then mounted on a Leica microtome (Leica biosystems, UK) and tissue sections of 10 μm were collected onto glass microscope slides. Tissues were subsequently stained using 1:400 dilution of primary antibody in PBS plus 0.4% Triton X-100 and 10% donkey serum. Antibodies used: SERCA2a (MA3-910, Thermo Fisher Scientific, USA) and anti-HA antibody C29F4 (Cell Signalling Technology, USA). The tissues were subsequently washed with blocking buffer and incubated with fluorescently labelled secondary antibodies. Nuclei were stained with Gold Anti-fade Reagent with DAPI (Invitrogen, USA). A confocal microscope (ZEISS, LSM 700) was used for imaging analysis.

**Preparation of heart lysates and western blot analysis.** For western blot analysis, the snap-frozen hearts were placed in a pestle and mortar and pulverised. The powdered tissue was distributed over six round bottom 2 ml eppendorf tubes. Six hundred microlitres of NP40 (NP40 Cell Lysis Buffer, Thermo Fisher Scientific, UK), Pierce protease, phosphatase inhibitor (Thermo Fisher Scientific, UK) was added to each tube along with a stainless steel metal bead. The tubes were placed in a bullet homogeniser and were homogenised twice at 50 Hz for 2 min in the cold room. The tubes were then left rotating in lysis buffer at 4 °C for another 2 h to continue lysis. After lysis the beads were removed and the lysed tissue was

centrifuged at $20,000 \times g$ for 20 min at 4 °C in a table-top eppendorf microcentrifuge. The supernatant was collected and the protein concentration was measured using the BCA method (Pierce™ BCA Protein Assay Kit, Thermo Fisher Scientific, UK). Protein samples were subsequently analysed in denatured and reduced conditions by western blot. Before imaging, the membranes were cut at the 30 kDa marker due to a strong non-specific band present in all samples around 50 kDa due to the anti-HA primary antibody staining (Source Data Fig. 4). Membranes evaluating the VHH2 B4B4-HA expression were developed using Pierce™ ECL Western Blotting Substrate (thermo Scientific, UK). Membranes evaluating VHH$_2$ C6C6-HA expression were developed using SuperSignal West Pico PLUS Chemiluminescent Substrate (Thermo Scientific, UK).

**Calcium transient measurements of adult cardiomyocytes.** Nine- to twelve-weeks-old C57BL/6 mice (Taconic, Denmark) were injected intracardially with 50 μg of modRNA encoding for VHH$_2$ B4B4-T2A-mCherry ($n = 3$), VHH$_2$ C6C6-T2A-mCherry ($n = 3$), and a negative control bivalent VHH$_2$ Cas9-T2A-mCherry ($n = 3$) formulated in RNAimax + Optimem (2:1:1; w:v:v) as described above. After 24 h, mice were sacrificed and the heart was quickly excised and subjected to Langendorff perfusion, essentially according to Louch et al.[71]. Briefly, the excised heart was immersed in 50 ml of ice-cold oxygenated isolation buffer with the addition of 1 ml of heparin (100 U vial, Leo Pharmaceutical Ltd, Denmark) and cannulated via the aorta on a 1 mm cannula and using a syringe filled with oxygenised isolation buffer with the added heparin. Immediately after cannulation, the coronary arteries were cleared by injecting the buffer via the syringe through the canula. The heart was then transferred to the Langendorff apparatus primed with warm (37 °C) isolation buffer as quickly as possible, and perfused for a few minutes at a constant flow rate of 3 ml/min. The buffer was then changed to isolation buffer with the addition of 2.0 mg/ml collagenase II (Worthington Biochemical Corporation, USA) and 0.01 mM CaCl$_2$. The digestion was allowed to proceed for 6 min at 3 ml/min. The digested tissue was removed from the Langendorff apparatus and the injection area of the heart was identified and isolated from the rest of the ventricles. Both tissues were then further processed by gentle separation and trituration. Ca$^{2+}$ tolerance of the cell prep was accomplished by moving the cell pellet into a fresh warm (37 °C) isolation medium at stepwise higher CaCl$_2$ concentrations (mM; 0.01, 0.1, 0.2, 0.3 and 0.5). At each step cells were incubated for 15 min to allow equilibration and sedimentation before moving to the next solution. The cell prep was left in the dark at room temperature and was used for the acquisition of calcium imaging data.

Fluo-4 AM powder (Thermo Fisher Scientific, UK) was dissolved in pluronic (Thermo Fisher Scientific, UK) to make a 1 mM stock solution. Five hundred microlitres of cell suspension was incubated with 5 μL of Fluo-4 AM for 20 min at room temperature and then washed twice by gentle resuspension and sedimentation for 5 min in Hepes buffered Tyrodes pH 7.4 containing 1 mM CaCl$_2$. Cells were then carefully resuspended in the Tyrodes buffer containing 1 mM CaCl$_2$, plated on mouse laminin (Thermo Fisher Scientific) coated Matek dishes (at 20 μg/ml) and imaged after 5 min to allow for sedimentation. Images were captured using a Nikon Eclipse wide-field microscope equipped with a CCD camera for high-speed acquisition (Andor Zyla 4.2 +, Oxford Instruments, UK).

The image stack was analysed using the Nikon elements analysis software and Fiji-Image J[70]. Each cell was manually picked and a region of interest (ROI) was contoured using the auto-detection tool, manual inspection and the time-resolved fluorescence intensities were measured. Fluo-4 fluorescence intensities were plotted as $(F − F_0)/F_0$ ($\Delta F/F_0$), where $F$ is the fluorescence intensity and $F_0$ is the fluorescence intensity at the diastole. Peak height, time to peak, $T_{50}$ and $\tau$ were measured and average values were calculated from three consecutive peaks. $\tau$ was determined by non-linear curve fitting of the relaxation phase of the Ca$^{2+}$ transient in Fiji-ImageJ. Cells with $\Delta F/F_{0\ max}$ below 100 RU and those with aberrant behaviour (aberrant pacing or the presence of extensive calcium waves or extensive spontaneous twitching) were rejected from the analysis. Cells with mCherry fluorescence above background levels were identified as transfected cells. Cells with no mCherry fluorescence above background levels in the sample were identified as non-transfected cells.

**AAV vector production.** AD293 cells (Agilent, 240085) were plated in a five-layered chamber (Corning, 3313) in DMEM (Gibco) supplemented with 10% FBS. Next day, cells were transfected by polyethylenimine (Polyscience, 24765-2) based triple plasmid transfection of pHelper containing adenoviral E2A and E4 genes, pRep2Cap9 encoding AAV2 Rep proteins and AAV9 serotype capsid, and either pAAV TNT455 ZsGreen or pAAV TNT455 ZsGreen PLN VHH$_2$B4B4 in 2:1:4:1 ratio respectively. After 72 h post-transfection, cells were harvested and lysed via three freeze-thaw cycles followed by benzonase treatment. Supernatants were then further purified using iodixanol gradient-based ultracentrifugation. Titers of all AAV vectors were measured by qPCR.

**In vivo delivery of AAV9 vectors in MLP KO mice.** Male muscle lim protein-deficient mice ($MLP^{−/−}$), 14–17 weeks of age, on black Swiss background, were purchased from Taconic (Denmark). $MLP^{−/−}$ mice show systolic dysfunction in terms of reduced ejection fraction, stroke volume and cardiac output.

$MLP^{−/−}$ mice were acclimatised for 5 days before the start of the study. The day before treatment started, a 2D transthoracic echocardiography (Vevo 2100 system

with 40 MHz transducer (FUJIFILM VisualSonics) was performed. Parasternal short-axis and long-axis views were obtained at Left Ventricular (LV) mid-papillary level, to assess ejection fraction (Vevo LAB software version 3.2.6 (FUJIFILM VisualSonics). Briefly, mice were first anaesthetised with a mixture of isofluorane 3–4% (Vetflurane®, Virbac, Suffolk, UK), air ~0.1 L/min and oxygen ~0.1 L/min in a gas chamber and then passed to a warm table (~38–40 °C) with a facemask that allowed a constant air/isofluorane (~2%) mixture throughout the echocardiography. The animal body temperature (~36–37 °C), respiration rate (50–100 rpm) and heart rate were monitored during all procedure. The animals with an EF ≤ 45% were included in the study. Mice were randomised into treatment groups based on bodyweight and EF values (≤45%). The next day, mice received a tail i.v injection of $10^{13}$/kg particles of AAV9 TNT455 ZsGreen ($n = 7$) or AAV9 TNT455 VHH$_2$ B4B4-T2A-ZsGreen ($n = 7$), in a volume of ~100 µl per animal. Bodyweight was monitored once weekly and hemodynamic measurements were performed before termination.

**Hemodynamic analysis**. At termination, the animals were anaesthetised with isofluorane (5%) (Attane vet, VM Pharma AB, Stockholm, Sweden) in a gas chamber and kept anaesthetised by spontaneously breathing isofluorane gas (2.75% through a mask). The gas concentration was established by exerting a flow of $O_2$ (0.5 L/min) and air (0.5 L/min) to a solution of isofluorane. To monitor the respiration rate, a pressure-sensitive pad was placed on the stomach of the mice. The pad was connected to a pressure sensor and the signal was recorded by LabChart (v8.1.16, ADInstruments, USA). Following surgery, the breathing rate was adjusted to about 40 breaths/min by adjusting the isofluorane concentration. An electrocardiogram (ECG) was recorded from skin electrodes. Hemodynamic parameters, $dP/dt_{max}$, $dP/dt_{min}$, LVPed, LVPes, heart rate and $\tau$ were measured via a 1-Fr pressure catheter (PVR-1035, Millar Instruments, Houston, Texas, US). The catheter was inserted into the left ventricle via the right carotid artery and hemodynamics was measured via the MPVS-Ultra Single Segment Foundation System (Millar, ADInstruments, US). The body temperature was maintained at 37 °C by external heating both from the table and from a heating lamp during the experiment. After a 10–20 min stabilisation period, baseline cardiac function was recorded. All parameters were calculated as a mean value over 10 s.

**Western blot analysis of MLP KO tissue samples**. Eighty milligrams of heart (base), quadriceps and liver tissue were collected from all MLP$^{-/-}$ mice included in the study ($n = 7$ (AAV9 TNT455 VHH$_2$ B4B4-T2A-ZsGreen); $n = 7$ (AAV9 TNT455 ZsGreen)). Forty milligrams of the tissue was homogenised in 250 µl RIPA buffer (50 mM Tris, 150 mM NaCl, 1% Sodium deoxycholate, 1% Sodium lauryl sulphate, 1% Triton-X-100 with Pierce protease, phosphatase inhibitor (Thermo Fisher Scientific, UK) using a metallic bead tissue lyzer for 2 min at 25 Hz. An additional 250 µl of RIPA buffer was added to the homogenised tissue and incubated for 1–2 h at 4 °C, before the samples were centrifuged at $16,000 \times g$ for 20 min at 4 °C in a table-top eppendorf microcentrifuge. Supernatants were used for BCA quantification (Pierce™ BCA Protein Assay Kit, Thermo Fisher Scientific, UK). Western blots were carried out as described above. Pierce™ ECL Western Blotting Substrate (Thermo Scientific, UK) was used to evaluate VHH expression in the heart, whilst SuperSignal West Pico PLUS Chemiluminescent Substrate (Thermo Scientific, UK) was utilised to detect VHH expression in quadriceps and liver samples.

**Statistical analysis**. Distinct samples were used for all analyses. All data are represented as means ± standard error of the mean (SEM). Two-sample $t$-tests were used to compare two groups with normal distributions. In case of more than two group comparisons, one-way ANOVA with two-sided post-hoc Tukey-Kramer (comparisons of the means between groups) or two-sided Dunnett's tests (comparison of the mean of each group to that of the control group) were utilised. All analyses were carried out using GraphPad Prism software version 9.0 (GraphPad Software Inc.). $P$-values of <0.05 were considered statistically significant.

**Reporting summary**. Further information on research design is available in the Nature Research Reporting Summary linked to this article.

## Data availability
Data necessary to interpret, verify and extend the research are available in the main article file, the supplementary materials and the Source data file. All other raw data can be obtained from the corresponding authors upon reasonable request. Source data are provided with this paper.

## Code availability
The Ca$^{2+}$ peak analysis program P.A.P was deposited as open-source code: https://github.com/xidan21/PAP.

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

## Acknowledgements

We would like to thank Elif Eroglu, Niels Grote-Beverborg, and Andrew Hall for valuable scientific discussions. Tamsin Albery, Nina Krutrök, Patricia Rodrigùes and Leif Aasehaug for technical assistance. We thank scientists at Isogenica for reviewing critically the manuscript. Live cell calcium imaging was performed at the Live Cell Imaging unit/Nikon Center of Excellence, BioNut, KI, supported by Knut and Alice Wallenberg Foundation, Swedish Research Council, Centre for Innovative Medicine and the Jonasson donation. This work was supported by grants from the European Research Council (ERC) under the European Union's Horizon 2020 research and innovation programme to K.R.C. grant agreement No. 745225, and Swedish Research Council Distinguish Professor Grant no. 541-2013-8351 to K.R.C.

## Author contributions

The study was initially conceived by K.R.C. with input from J.H. L.M.M. and E.D.G. E.D.G. wrote the manuscript, performed biopanning, VHH isolation and biophysical characterisation and protein engineering. E.d.V. constructed the VHH phage-display library. E.D.G. and N.W. designed ModRNA constructs. E.D.G., N.W. and J.S. prepared modRNA materials. E.D.G. and E.R. performed expression and target engagement assays in cell models including rat neonatal cardiomyocytes. E.D.G. and A.H. performed and analysed $Ca^{2+}$ imaging experiments in rat neonatal cardiomyocytes. X.L. wrote the computer code for the analysis of rat neonatal cardiomyocyte $Ca^{2+}$ flux analysis. Langendorff cardiomyocyte Isolation E.D.G. and Y.X., under supervision of W.E.L. and T.K. Calcium imaging and analysis of isolated cardiomyocytes E.D.G. and W.E.L. Design and murine intracardial injections, K.S.F. and Y.X. Immunofluorecence imaging K.S.F., J.S. and E.D.G. Heart lysates westernblotting, E.D.G. and E.R. AAV vector design and production, A.P. and Y.I. K.J., K.H., E.D.G., K.R.C., Y.I. and K.S.F. design of in vivo AAV transduction experiments. S.P. performed heart hemodynamic measurements, S.P., K.J. and K.H. analysis. K.S.F., J.H. and K.R.C. edited the paper. K.R.C., R.F.D., K.W., L.M.M., J.H. and E.D.G. design of the project. All authors discussed the results and reviewed critically the paper.

## Funding

## Competing interests

E.D.G., E.d.V., K.W., S.P., A.P., Y.I., K.J., K.M.H., R.F.D. and J.H. are employees and stockholders of AstraZeneca. L.M.M. has been a former employee at AstraZeneca and is a current stockholder of AstraZeneca. E.D.G. and A.H. were funded as post-doctoral fellows by the AZ innovation fund. The remaining authors declare no competing interests.
