## [Peer Review File · Nature Communications]

REVIEWER COMMENTS

Reviewer #1 (Remarks to the Author):

Title: Blocking phospholamban with VHH intrabodies enhances contractility and relaxation in heart failure

Manuscript Number: NCOMMS-21-27361

The authors show the first application of VHH intracellular acting antibodies (intrabodies) to block different conformational states of phospholamban (PLB). Dysregulation of calcium handling processes in heart failure is well described and there is an urgent need to treat this condition. The authors developed a new phage-display library to generate intrabodies with high specificity to different conformational states of PLB. Intrabodies were delivered via modRNA in vitro and in vivo and correct localization and functional effects on calcium cycling were ensured in different cardiomyocyte lines. Ultimately, improvement of contractile function could be confirmed in MLP-KO mice with diastolic heart failure. This study is well suited for publication in the journal. However, major revision is required to improve the quality of the manuscript. Especially a better presentation of the figures is needed.

Major comments:

- 1) The authors transduced the MLP-KO mice with AAV9 vectors encoding the intrabody VHH2 B4B4 and could show an improvement of in vivo cardiac contractility and faster relaxation parameters of contraction. For cardiac function, it would be nice to show also the basal mice without KO phenotype (non-transduced and transduced with the AAV). To which extent does treatment reverse the impairment of the relaxation parameters as compared to basal phenotype?
- 2) Could the authors include a timeline for viral expression, intrabody expression and contractility parameters over time, longer than 14 days after transduction to show how long increased expression levels and functional improvement persist?
- 3) Please add single cell contractile measurements to prove improvement of contractility and relaxation after modRNA transfection or viral transduction of the intrabodies.
- 4) In figure 3i it is not clear why PLB fluorescence accumulates perinuclear. Please provide staining with another specific PLB/SERCA antibody.
- 5) For figure 3e-h, the transfected control group should be used and not the untransfected one.
- 6) In order to verify that there is no toxicity, an MTT assay should be performed in vitro.
- 7) Please add more samples for figure 4b,c to make results more reliable.

8) According to the protein expression of α -actinin is very low (supplemental figure 6, left panel). Why is it the case? Use alternatively GAPDH to repeat the experiments.

9) To prove the efficiency of VHH C6 please perform additional experiments upon adrenergic stimulation of cardiomyocytes (to increase the fraction of phosphorylated PLB) and do calcium imaging experiments on treated control group and with VHH C6 transfected group.

10) In supplemental figure 6, the quality of WBs is bad. Please improve the quality of the WBs.

11) Please include a section on the performed statistical analysis and also indicate in each figure legend which statistical parameters (e.g. mean, median values and the respective errors etc.) are shown.

Minor comments:

1) From the logic of the original text, please revise the order in figure 1b. The graph to VHH C6 should be presented before the graph to VHH2B4B4.

3) In figure 2b, lower panel, the lanes should be aligned with the labels.

4) Please make clear what is the difference in figure 2c between the middle and lower panel. Please label it clearly to visualize the differences. The size of the marker bands should be included also in the main figure 2b,c.

5) Please include into methods part differentiation and cultivation procedures of hESC- and iPSC-cardiomyocytes.

6) In Figure 3a-d, please label the representative traces.

7) Please adjust y-axis for 3a-d to the same value.

8) Please correct the unit at y-axis of graphs 3e to "ms".

9) Please exchange in figure 3i VHH C6 with a focused image and include the scale bars.

10) Please make clear which criteria you used to select the positive cells by arrows. The presumable perinuclear accumulation of both signals has different evaluation criteria for images in different columns of figure 3i.

11) In lines 211, please write 72 hours instead of 3 days to ensure unity.

12) There is no scale bar in Figure 4d, please add.

13) In figure 4a it is not yet clear which procedures were performed on the extracted hearts. In the text, it is stated that whole tissue was lysed for the WB, but it did not match the scheme, in which IF on the same tissue was also performed. Please combine instructions.

14) The size of the marker bands in figure 4b,c should be included.

15) Please adjust y-axis for 5b-e to the same value.

- 16) Please set x-axis in figure 5b-e to 5 sec. for each graph.
- 17) Please refer to figure 5b-e in the text before proceeding to figure 5f-h.
- 18) In order to indicate the level of significance, please select p-values or asterisks in all figures to ensure uniformity.
- 19) Please label the components in Figure 7 and describe the standard of the arrow in the legend.

Reviewer #2 (Remarks to the Author):

Muscle contractility is controlled by an array of Calcium handling proteins that make up the excitation-contraction (EC) coupling in cardiac cells. Among these proteins is the complex between phospholamban (PLN) and the sarcoendoplasmic reticulum Ca^{2+} -ATPase (SERCA), responsible for cardiac muscle relaxation. Decades of structural and functional studies on this complex have culminated in the recent structural model that see PLN undergoing significant conformational changes. The resting form is a pentameric assembly that dissociates in active monomers that bind and regulate SERCA reversibly. Phosphorylation by cAMP-dependent kinase is a critical event which reverses the inhibitory action of PLN. Genetic mapping has revealed a handful of deadly mutations of PLN that result in early or late onset of disease, which eventually leads to heart failure. Due to the membrane architecture of PLN, there are not apparent druggable sites, and most of the developed compounds are directed to SERCA. However, the latter is a significant complication as SERCA isoforms are widely distributed in many muscle tissues, and their high homology may determine significant countereffects for drugs. Antibodies can reverse the action of PLN, or as more recently reported, by single-stranded oligonucleotides (Soller et al. JBC 2016 and Soller et al. Sci Rep 2015). However, the delivery of these systems is quite hard. In this new and exciting paper, Chien and co-workers developed a novel approach to deliver intrabodies to reverse the inhibitory action of PLN selectively. Using co-IP, these scientists found specific sequences with high affinity for PLN (nM range) that can distinguish the different states of PLN (phosphorylated vs unphosphorylated). The possibility to direct intrabodies to specific states of this essential mini membrane protein is vital. In particular, Chien shows a high affinity for the pentameric state of PLN, which represents the reservoir for active monomers. In addition to developing these sequences, Chien and co-workers also show that it is possible to easily deliver these intrabodies into cardiomyocytes obtaining only marginal effects on the amplitude of the Ca^{2+} transient but significant effects on the kinetics of relaxation. The ultimate test was to deliver these intrabodies in mice models for dilated cardiomyopathy to see whether it could reverse the dysfunctional contractility. The data show that indeed the intrabodies localize in the SR of the cardiac muscle and target PLN directly (see Figure 4). The most exciting part of this study is the correlation between the physiological and molecular biology work with the most recent structural models that have been published. The latter enabled these scientists to fine-tune the interfering intrabodies to the different states of PLN, and in

particular, target either phosphorylated or unphosphorylated pentamer, further supporting the previous studies.

For most of the parts, the paper is well written and suitable to the broad audience of your journal. There are a few minor comments that the authors need to address before publication.

A) The model in figure 7 (top panel) should be complete with two arrows completing the cycle. It has been shown that monomeric PLN bound to SERCA interchange between the T and B state, helical and extended structure, respectively (Gustavsson et al. 2003).

B) Did the authors try to shorten the intrabodies to identify a minimum binding site?

C) The authors should add more recent references regarding PLN deadly mutants such as R9C, R25C, R14del, L39stop, RL9, RH9. See, for instance, Ha et al. PNAS 2011

D) Did the authors try this approach with mice or cardiomyocytes carrying R14del mutation?

E) How does this possible approach compare with the recent correction of PLN R14del proposed by Kranias and Hajjar (Nat Comm 2015)?

Reviewer #3 (Remarks to the Author):

My summary review is that the ms is very good, and breaks new ground. Methods and data analysis are sound, with sufficient detail for the antibody work. It is acceptable with minor modifications in the discussion.

The dimerization to increase avidity and target the pentamer is especially impressive.

The modRNA worked well for the very short-term experiments, although it is not clear that it is going to perform better than a DNA plasmid when the latter can be delivered efficiently in the chosen cell line in many circumstances.

The discussion is already quite detailed. One issue that should also be considered --

When delivering the intrabodies systemically, what is the chance of an off-target effect? Are the screening peptides shared with any other proteins? How tight is the control afforded by the promoter that was used with the AAV9, since AAV9 will also go to the brain?

Blocking phospholamban with VHH intrabodies enhances contractility and relaxation in heart failure

We would like to thank the reviewers for the critical, insightful comments, particularly pointing out the novelty and broad interest of the work. We are appreciative of the reviewers for their constructive comments, which have helped us to resubmit an improved version of the original manuscript. In accordance with the suggestion of the reviewers, we have decided to expand our initial study and integrated new data into a well revised version of the original submission, which have strengthened the data presented in the original study. Regarding the new data collected after the original submission (please see Appendix Table 1 for a complete list of revisions made). In addition, please find below is a detailed response to each point raised by the reviewers.

Reviewer 1: *The authors show the first application of VHH intracellular acting antibodies (intrabodies) to block different conformational states of phospholamban (PLB). Dysregulation of calcium handling processes in heart failure is well described and there is an urgent need to treat this condition. The authors developed a new phage-display library to generate intrabodies with high specificity to different conformational states of PLB. Intrabodies were delivered via modRNA in vitro and in vivo and correct localization and functional effects on calcium cycling were ensured in different cardiomyocyte' lines. Ultimately, improvement of contractile function could be confirmed in MLP-KO mice with diastolic heart failure. This study is well suited for publication in the journal. However, major revision is required to improve the quality of the manuscript. Especially a better presentation of the figures.*

Major comments:

1) The authors transduced the MLP-KO mice with AAV9 vectors encoding the intrabody VHH2 B4B4 and could show an improvement of in vivo cardiac contractility and faster relaxation parameters of contraction. For cardiac function, it would be nice to show also the basal mice without KO phenotype (non-transduced and transduced with the AAV). To which extent does treatment reverse the impairment of the relaxation parameters as compared to basal phenotype?

We thank the reviewer for this suggestion and agree that it is important to understand the impact of intrabody expression on cardiac function in the normal heart. However, the wild-type mouse heart works at nearly maximal capacity, therefore the added physiological benefit of inhibiting PLN function by intrabody expression will be difficult to detect statistically due to biological variation and variability in intrabody expression. In the MLP KO heart failure model, cardiac function is sufficiently impaired to detect a significant improvement in cardiac function by intrabody inhibition of PLN using a reasonable number of animals. Instead, we feel that the suggested experiment is captured adequately and robustly in our *in vitro* Ca²⁺ measurements showing that VHH₂ B4B4 intrabody expression in adult wild-type mouse cardiomyocytes results in enhanced calcium cycling. These results are less biased by biological variation than physiological experiments and therefore a more robust measure of the effects on the baseline wild-type state. These *in vitro* experiments formed the

basis for our decision to pursue the measurements of the physiological effects of VHH₂ B4B4 *in vivo* using a murine heart failure model.

The main aim of our *in vivo* study was to provide a proof of concept to our intrabody strategy, from synthetic library selection to physiological readout, with potential therapeutic effects in a disease model. The comparison with WT phenotype at the physiological level in combination with a more in depth pharmaco-kinetic and pharmaco-dynamic evaluation of VHH₂ B4B4 lies beyond the scope of this paper and would be better suited in a preclinical translational study.

2) Could the authors include a timeline for viral expression, intrabody expression and contractility parameters over time, longer than 14 days after transduction to show how long increased expression levels and functional improvement persist?

Previously, we employed an AAV9 vector and demonstrated long-term heart-targeted expression of a non-immunogenic protein at least for 1.5 years in rodents (Cataliotti *et al.*, 2011 *Circulation*; Tonne *et al.*, 2015 *Aging*).^{1,2} On the other hand, we also found that the cardiac-targeted AAV9 vector delivery can induce cellular and humoral immune responses at three weeks post injection when an immunogenic transgene product was expressed in the heart. (Please see the provided confidential data, recently presented at the annual American Society for Gene and Cell Therapy meeting). Therefore, although we agree that it would be interesting to evaluate later time points for the *in vivo* expression, we based our decision to limit the study to 14 days on this observation, since our intrabodies are foreign proteins, which could potentially induce adaptive immune responses and affect cardiac functions. For clinical applications of AAV delivered intrabodies, we would de-immunize our intrabody sequences to minimize potential risks of transgene immunogenicity, which would be within the scope of a preclinical follow-up study. To emphasize this point we have added, the following sentence to the main text of the manuscript including the references mentioned above: page 11 line 306: “We chose a transduction period of 14 days as we have previously shown that AAV9 transgene expression is maximal after this time, whilst minimising the risk of an adverse immune response against the non-endogenous protein VHH₂ B4B4-T2A-ZsGreen.”

1. Cataliotti, A. *et al.* Long-term cardiac pro-B-type natriuretic peptide gene delivery prevents the development of hypertensive heart disease in spontaneously hypertensive rats. *Circulation* **123**, 1297–1305 (2011).
2. Tonne, J. M. *et al.* Cardiac BNP gene delivery prolongs survival in aged spontaneously hypertensive rats with overt hypertensive heart disease. *Aging (Albany NY)* **6**, 311–319 (2014).

3) Please add single cell contractile measurements to prove improvement of contractility and relaxation after modRNA transfection or viral transduction of the intrabodies.

We thank the reviewer for this suggestion. We have not assessed cell shortening/relaxation, as considerable previous work has shown that cardiomyocyte calcium transients and contraction/relaxation measurements show parallel changes when SERCA2a activity is modulated. For example, in the early work of Del Monte *et al* (*Circulation*, 2002)³ they observed that phospholamban antisense or SERCA overexpression had very similar effects on the rate of Ca²⁺ reuptake and lusitropy:

Figure adapted from Del Monte et al, *Circulation* 2002, showing a clear correlation between cell contraction-relaxation and SERCA2a modulation, reflected both in the calcium transient and the cell shortening.

We expect that is also the case during SERCA modulation in the current study, as *in vivo* functional changes mirrored alterations in calcium homeostasis measured *ex vivo*. We have therefore concentrated our efforts on other experiments during the revision process.

- del Monte, F., Harding, S. E., Dec, G. W., Gwathmey, J. K. & Hajjar, R. J. Targeting phospholamban by gene transfer in human heart failure. *Circulation* **105**, 904–907 (2002).

4) In figure 3i it is not clear why PLB fluorescence accumulates perinuclear. Please provide staining with another specific PLB/SERCA antibody.

The accumulation of PLN staining at the perinuclear region in rat neonatal cardiomyocytes or other immature models, such as iPS derived cardiomyocytes cultured *in vitro*, is well understood as the sarcoplasmic reticulum (SR) is attached to and for the larger part located near the nuclear envelope. The apparent immature organisation of the SR in these cells therefore leads to a more pronounced perinuclear staining with PLN specific antibodies in contrast to a more striated staining pattern of the SR observed in mature cardiomyocytes. This staining pattern can also be observed when iPS derived cardiomyocytes are transfected with tagged PLN constructs and stained using anti-tag antibodies, as we have recently demonstrated (Rohner et al, 2021, BMC)⁴. Furthermore, PLN has also been found to be expressed in the nuclear membrane itself (Chen et al, *JMCC* 2018; He et al, *J Mol Cell Cardiol* 2020)^{5,6}.

The 2D12 antibody is also widely used and verified in the field and considered to be a gold standard antibody for PLN detection and has been referenced in over 140 peer reviewed publications. To provide further validation for the anti-PLN 2D12 antibody in immunofluorescence studies, we have now included staining of heart tissue containing mature cardiomyocytes, showing a striated pattern characteristic for the organisation of the SR found in mature cardiomyocytes. We have added this figure in the supplemental information as “*Supplementary Figure 10*” for additional validation for the 2D12 antibody in immunofluorescence staining. We also refer to this in the methods section, p 45 line 1079: “*The use of the anti-PLN antibody 2D12 for immunofluorescence studies was validated using adult murine heart tissue (Supplementary Fig. 10)*”.

- Rohner, E. et al. An mRNA assay system demonstrates proteasomal-specific degradation contributes to cardiomyopathic phospholamban null mutation. *Mol Med* **27**, 102 (2021).
- Chen, M. et al. Phospholamban regulates nuclear Ca²⁺ stores and inositol 1,4,5-trisphosphate mediated nuclear Ca²⁺ cycling in cardiomyocytes. *J Mol Cell Cardiol* **123**, 185–197 (2018).
- He, W. et al. Association with SERCA2a directs phospholamban trafficking to sarcoplasmic reticulum from a nuclear envelope pool. *J Mol Cell Cardiol* **143**, 107–119 (2020).

5) For figure 3e-h, the transfected control group should be used and not the untransfected one.

We agree with the reviewer that the transfection of a non-relevant intrabody would have been ideal as a control, as opposed to untransfected cells. Nevertheless, the rat neonatal cardiomyocyte assay was intended to triage the intrabody candidates to be taken forward in the physiologically more relevant adult cardiomyocytes and *in vivo* models, where two excellent controls thoroughly validated the observed effects. However, to accommodate the reviewer's concern, we have included new data originating from rat neonatal cardiomyocytes transfected with the modRNA constructs depicted in figure 5, which code for the intrabodies fused to a T2A peptide and the mCherry reporter protein, in comparison with data from untransfected cells originating from the same batch of isolated cells. Transfected cells were identified by measuring the red mCherry fluorescence and the calcium cycling parameters were measured using Fluo-4 fluorescence measurements. The Ca^{2+} transient amplitude, T_{50} , upslope velocity and the relation rate constant $1/\tau$ were compared to untransfected cells. Although a smaller sample set of cells was used in this experiment compared to those in figure 3e-h, no significant differences between untransfected or VHH₂ Cas9 transfected were observed, other than a small but significant difference in the peak amplitude. Identical trends were observed for VHH₂ B4B4 and VHH₂ C6C6 compared to Fig 3e-h. We have included this new data in the supplementary materials as "Supplementary Figure 7" and we have referred to this data in the main text, p 8, line 234: "*Transfections of these new constructs in cultured rat neonatal cardiomyocytes resulted in similar observations to those made for non-fluorescently tagged intrabody constructs (Supplementary Fig. 7). A small difference in the Ca^{2+} peak amplitude was however observed between the negative control VHH₂ Cas9-T2A-mCherry and untransfected rat neonatal cardiomyocytes, underscoring the importance to include this non-relevant transfection control.*"

6) In order to verify that there is no toxicity, an MTT assay should be performed in vitro.

We acknowledge the reviewer's concern, and we have in fact verified cell toxicity of modRNA transfection using a Cell Titer Glo viability assay from Promega, which is a homogeneous method of determining the number of viable cells in culture based on quantitation of the ATP present, an indicator of metabolically active cells. We performed this assay on iPS derived cardiomyocytes transfected with modRNA encoding for VHH B4 and VHH C6 as well as a negative control VHH, VHH Cas9 early on in the research study. The experiments show no observable cytotoxicity in iPS derived cardiomyocytes. This data was not originally present in the manuscript but we have added this new data as "Supplementary Figure 4 a", in the supplementary materials and referred to these results in the main manuscript on p 5 line 143: "*Transfections of modRNA encoded intrabodies in different formats, including, monomeric, bivalent or as genetic fusions with fluorescent proteins, showed furthermore no detectable cytotoxicity (Supplementary Fig 4 a,b).*" Additionally, we have also assessed the potential cytotoxic effects due to transfection of iPS derived cardiomyocytes with the modRNA constructs VHH₂ B4B4 and VHH₂ C6C6, as well as T2A mCherry genetic fusions VHH₂ B4B4 T2a mCherry, VHH₂ C6C6 T2a mCherry and VHH₂ Cas9 T2a mCherry and similar to the monovalent constructs, we have not found any indication for cytotoxicity within the time frame of the experiments reported in this study. We have included this new data in the supplementary materials as "Supplementary Fig 4b". Furthermore, in our extensive *in vitro* and *in vivo* experiments, using both modRNA transfection and AAV9 transduction, we have not observed any indication of cell toxicity in any of the constructs. These results are in line with other studies using VHHs as intrabodies and again underscore their suitability as an intracellular acting platform.

7) Please add more samples for figure 4 b,c to make results more reliable.

We thank the reviewer for raising this concern, along with “minor comment #13”. We believe that this concern stemmed from a lack of clarity regarding the procedure depicted in Figure 4 a, for which we apologise. We would like to emphasize and clarify that different mice were used to perform the immunofluorescence experiments and western blots, therefore resulting in a total sample number of 6 mouse hearts per intrabody, in contrast to the suggested procedure depicted in the original figure. The western blot samples were generated from whole heart lysates and three independent experiments are shown on the blots, which we believe is an adequate representation and transparent evaluation of the data. The difference in expression reflects biological and procedural variation (injection procedure and transfection efficiency). It is however not our aim to provide any quantitative assessment of the expression levels, but merely qualitative assessment and demonstration of the expression kinetics of the modRNA injected in the *in vivo* murine heart setting. We do recognise that the image in Fig. 4c does not show the expression of VHH₂ C6C6 very clearly. Therefore, we have exchanged the western blot in Figure 4 c, with one that was developed with a more sensitive HRP substrate, which enhanced the signal significantly. We have also added full-scale blots to the supplementary materials as Supplementary Fig. 7. We refer to this figure in the legend of Fig. 4 of the main text.

8) According to the protein expression of α -actinin is very low (supplemental figure 6, left panel). Why is it the case? Use alternatively GAPDH to repeat the experiments.

We agree with the reviewer, and we have repeated the western blots, using GAPDH as a loading control. We have replaced the original Figure 6 left panel with an improved blot. This figure is now referred to as Supplementary Figure 9 a. In addition, we have significantly improved the quality of the other western blot in this figure and also included new western blotting data for tissue samples originating from liver and quadriceps of the MLP KO mice used in this study. We have additionally adapted Figure 6 in the main text to include this improved data as well as the new data (Fig. 6 f,g).

9) To prove the efficiency of VHH C6 please perform additional experiments upon adrenergic stimulation of cardiomyocytes (to increase the fraction of phosphorylated PLB) and do calcium imaging experiments on treated control group and with VHH C6 transfected group.

This is a valid concern from the reviewer. Our intrabody VHH B4 and VHH₂ B4B4 fulfil the criteria for further exploration of their disease modifying potential, and therefore we chose to mainly focus on these intrabodies, whereas VHH C6 and VHH₂ C6C6 would have a less significant and opposite effects on the calcium cycling properties compared to VHH B4/VHH₂ B4B4. This is in agreement with a mechanism whereby VHH C6 targets a different of PLN species to VHH B4 and consistent with the model that is depicted in figure 7. We agree that an in-depth study on the mechanism of action of VHH C6/VHH₂ C6C6 will involve the use of catecholamines that stimulate PLN phosphorylation. However, due to the complexity of the phospho-signalling on calcium cycling in cardiomyocytes of different maturity, and the complex interaction of different populations of PLN and their interaction with VHH C6 or VHH₂ C6C6 will require a more extensive study, which is beyond the scope of this paper and will be explored in a future report. We have added a sentence in the discussion section to highlight the more speculative mode of action of this intrabody and the need for a more detailed investigation into the precise mechanism by which it influences Ca²⁺ signalling in cardiomyocytes using catecholamines and different cell models: p15 line 434: “*The expression of the phospho-specific intrabodies VHH C6 and VHH₂ C6C6 resulted in only minor effects on cardiomyocyte contractility and relaxation, leading to a more speculative model for the mode of action of these intrabodies.*” and, p

15 line 449: *“An in depth analysis of the Ca²⁺ dynamics in cardiomyocytes transfected with VHH C6 and VHH₂ C6C6, using catecholamines to modulate the phosphorylation state of PLN, will be required to provide further validation of this model.”*

10) In supplemental figure 6, the quality of WBs is bad. Please improve the quality of the WBs.

We agree with the reviewer, and we have improved the quality of the western blots and replaced the original blots. Please find improved versions of the western blots in Supplementary Figure 9 a,b. As mentioned in point 9, we have also added additional western blot data of tissue samples originating from the liver and quadriceps of the MLP KO mice in the study and we now presented this data in Figure 6 f,g in the main text.

11) Please include a section on the performed statistical analysis and also indicate in each figure legend which statistical parameters (e.g. mean, median values and the respective errors etc.) are shown.

We have adapted the figures and included a separate section on the description of the statistical analysis in the methods section under the heading “Statistical analysis” on p 50, line 1225 of the revised manuscript.

Minor comments:

1) From the logic of the original text, please revise the order in figure 1b. The graph to VHH C6 should be presented before the graph to VHH2B4B4.

We agree with the reviewer that the suggested reorganisation of the figure panels is more consistent with the flow of the text and we have swapped panels 1b and 1c in the revised figure.

3) In figure 2b, lower panel, the lanes should be aligned with the labels.

We thank the reviewer for spotting this error and we have now corrected this in the revised figure 2b.

4) Please make clear what is the difference in figure 2c between the middle and lower panel. Please label it clearly to visualize the differences. The size of the marker bands should be included also in the main figure 2 b,c.

We have clarified the differences between both panels in figure 2 c and we have included the molecular sizes marker bands.

5) Please include into methods part differentiation and cultivation procedures of hESC- and iPSC-cardiomyocytes.

We have moved the description of mammalian cell culture and transfection from the supplementary information to the methods section in the main manuscript (p 41, line 944)

Within this section, we have added the cultivation of the hESC cells: p 41 line 948: *“ES03 hESCs were cultured in E8 basal media (Thermo Scientific) and passaged every 3–4 days and plated on tissue culture plates coated with matrigel (Thermo Scientific).”*

In this section we have also added the description of the iPS derived cardiomyocyte culturing procedure. These iPS derived cardiomyocytes were purchased as differentiated cardiomyocytes and used as such and no further maturation of these cells to a more mature phenotype was attempted. On p 41 line 950: *“Human iPS derived cardiomyocytes were purchased from FUJIFILM Cellular*

Dynamics, Inc (FCDI), and frozen cells were thawed and plated according to the manufactures' recommendations."

6) In Figure 3a-d, please label the representative traces.

We have now labelled the traces with identity of the corresponding intrabody.

7) Please adjust y-axis for 3a-d to the same value.

We have now adjusted the panels in figure 3a-d to the same value

8) Please correct the unit at y-axis of graphs 3e to "ms".

We thank the reviewer for spotting this this typographical error, which has been corrected in the revised version.

9) Please exchange in figure 3i VHC6 with a focused image and include the scale bars.

We apologise for the unfocused image. We have adapted all the panels to be a composite of the focused plane of each channel, adjusted the brightness threshold to make the images more visible. Furthermore, we have ensured that scale bars are on all the IF images.

10) Please make clear which criteria you used to select the positive cells by arrows. The presumable perinuclear accumulation of both signals has different evaluation criteria for images in different columns of figure 3i.

We have now clarified in the figure legend that the arrows indicate cells that show clear co-localisation of PLN and HA signals including a more marked staining in the perinuclear region, typical of PLN expression.

11) In lines 211, please write 72 hours instead of 3 days to ensure unity.

This has now been corrected.

12) There is no scale bar in Figure 4d, please add.

We apologise for the oversight, scale bars are now added to the figures (main document and supplementary file) displaying microscopy images.

13) In figure 4a it is not yet clear which procedures were performed on the extracted hearts. In the text, it is stated that whole tissue was lysed for the WB, but it did not match the scheme, in which IF on the same tissue was also performed. Please combine instructions.

Please also refer to our response to Major comment #7. The figure is now adapted to clarify that different hearts were taken for IF staining and western blot.

14) The size of the marker bands in figure 4 b,c should be included.

This has been corrected.

15) Please adjust y-axis for 5b-e to the same value.

We have adapted the figure to accommodate this suggestion.

16) Please set x-axis in figure 5b-e to 5 sec. for each graph.

This has been corrected in the new version.

17) Please refer to figure 5b-e in the text before proceeding to figure 5f-h.

We thank the reviewer for spotting this error. We have now added a sentence on p 9 line 244: “Calcium flux recordings were subsequently performed using electrical pacing at 1Hz and 4Hz (Fig. 5 b-e). We found ...”

18) In order to indicate the level of significance, please select p-values or asterisks in all figures to ensure uniformity.

This has been corrected in the new version of the manuscript. We have used asterisks in the figure and have given the *P*-values in the text and in the supplementary materials.

19) Please label the components in Figure 7 and describe the standard of the arrow in the legend.

We have labelled the panels for each reaction and we have clarified the meaning of the different arrows in the figure legend. In addition we have swapped the order of the depictions of the mechanisms for the different intrabodies to make it consistent with the order they are described in the text. In addition some minor changes have been made, including references to the different panels in Fig. 7.

Reviewer #2 (Remarks to the Author): *Muscle contractility is controlled by an array of Calcium handling proteins that make up the excitation-contraction (EC) coupling in cardiac cells. Among these proteins is the complex between phospholamban (PLN) and the sarcoendoplasmic reticulum Ca²⁺-ATPase (SERCA), responsible for cardiac muscle relaxation. Decades of structural and functional studies on this complex have culminated in the recent structural model that see PLN undergoing significant conformational changes. The resting form is a pentameric assembly that dissociates in active monomers that bind and regulate SERCA reversibly. Phosphorylation by cAMP-dependent kinase is a critical event which reverses the inhibitory action of PLN. Genetic mapping has revealed a handful of deadly mutations of PLN that result in early or late onset of disease, which eventually leads to heart failure. Due to the membrane architecture of PLN, there are not apparent druggable sites, and most of the developed compounds are directed to SERCA. However, the latter is a significant complication as SERCA isoforms are widely distributed in many muscle tissues, and their high homology may determine significant counter effects for drugs. Antibodies can reverse the action of PLN, or as more recently reported, by single-stranded oligonucleotides (Soller et al. JBC 2016 and Soller et al. Sci Rep 2015). However, the delivery of these systems is quite hard. In this new and exciting paper, Chien and co-workers developed a novel approach to deliver intrabodies to reverse the inhibitory action of PLN selectively. Using co-IP, these scientists found specific sequences with high affinity for PLN (nM range) that can distinguish the different states of PLN (phosphorylated vs unphosphorylated). The possibility to direct intrabodies to specific states of this essential mini membrane protein is vital. In particular, Chien shows a high affinity for the pentameric state of PLN, which represents the reservoir for active monomers. In addition to developing these sequences, Chien and co-workers also show that it is possible to easily deliver these intrabodies into cardiomyocytes obtaining only marginal effects on the amplitude of the Ca²⁺ transient but significant effects on the kinetics of relaxation. The ultimate test was to deliver these intrabodies in mice models for dilated cardiomyopathy to see whether it could reverse the dysfunctional contractility. The data show that indeed the intrabodies localize in the SR of the cardiac muscle and target PLN directly (see Figure 4). The most exciting part of this study is the correlation between the physiological and molecular biology work with the most recent structural models that have been published. The latter enabled these scientists to fine-tune the interfering intrabodies to the different states of PLN, and in particular, target either phosphorylated or unphosphorylated pentamer, further supporting the previous studies.*

For most of the parts, the paper is well written and suitable to the broad audience of your journal. There are a few minor comments that the authors need to address before publication.

A) The model in figure 7 (top panel) should be complete with two arrows completing the cycle. It has been shown that monomeric PLN bound to SERCA interchange between the T and B state, helical and extended structure, respectively (Gustavsson et al. 2003).

We agree with the reviewer that PLN is in dynamic exchange with a T and B state and that phosphorylation shifts the equilibrium of the regulatory domain to the B-state which enables the interaction of this domain with a regulatory site on SERCA2a. However, only when the B state is phosphorylated, the resulting productive binding allows SERCA2a to be activated. The dynamic exchange between the T and B state is however important for the accessibility of the PKA enzyme. We did not however depict this dynamic exchange of the different states in the figure for clarity reasons, as we wished to focus on the main conformations in either the phosphorylated and non-phosphorylated states of PLN.

B) Did the authors try to shorten the intrabodies to identify a minimum binding site?

Our intrabody molecules are VHH domains, which derived from camelid heavy-chain antibodies. The VHH is a 10-15 kDa protein and is the minimal intact binding unit of an immunoglobulin antibody molecule. The structural integrity of the complete VHH domain is required for the functionality of the combining site formed by three hypervariable loops embedded in a stable three-dimensional scaffold. Reducing the size of this domain will induce unfolding and aggregation of the VHH. Thus, we have not attempted to reduce the size of the intrabodies as this will severely impact the functionality of the intrabodies, both in terms of stability, binding affinity and may additionally lead to cell toxicity due to aggregation induced proteostatic stress.

C) The authors should add more recent references regarding PLN deadly mutants such as R9C, R25C, R14del, L39stop, RL9, RH9. See, for instance, Ha et al. PNAS 2011

We thank the reviewer for the suggestion, and we have now added more references to highlight the pathology associated with a range of different PLN variants. We have added the following sentence in the text p 2 line 40: *“In addition, several hereditary pathological variants of PLN have been directly linked to dilated cardiomyopathy, including the variants R9C^{5,6}, R9L/H⁷, R25C⁸, R14del⁹, and a truncation mutant L39X.¹⁰”*

D) Did the authors try this approach with mice or cardiomyocytes carrying R14del mutation?

We thank the reviewer for raising this interesting question and we agree that the R14del mouse model would be an exciting system to evaluate the potential of PLN specific intrabodies to target this deadly mutant. Moreover, our approach could allow a deeper understanding of PLN R14del pathology, for which both the precise pathophysiological classification and the molecular mechanism of the disease are still very much a matter of debate in the field. However, our PLN specific intrabodies are only very weakly reactive with the R14del variant, underscoring the very high specificity of the VHH single-domain antibodies and our phage-display campaign. A new panning campaign or extensive screening of the existing selection for binders that are specific for the R14del mutation or cross react with this mutant will need to be performed first before attempting to apply our strategy to this heart failure model, which we are planning to perform in a future study.

E) How does this possible approach compare with the recent correction of PLN R14del proposed by Kranias and Hajjar (Nat Comm 2015)?

We thank the reviewer for highlighting this work. Indeed, in the cited report, a gene silencing approach was used to knock down endogenous PLN & PLNR14del expression in patient derived iPSC CM that carried the c.40_42delAGA heterozygous mutation in the PLN gene. The authors showed that the mutation resulted in Ca²⁺ handling abnormalities, which they were able to reverse the disease phenotype by reducing endogenous PLN & PLNR14del. The authors achieved this using specific PLN inhibitory miRNA and providing miRNA resistant WT PLN sequence to correct for this mutation, although a lesser but significant drop of endogenous PLN was also observed when a non-relevant miRNA was used, underscoring the tight regulation of PLN expression. Thus, in this approach both endogenous PLN and the PLNR14del population are decreased and substituted by WT “healthy” PLN at the gene level. There are indeed other valid ways to target PLN mediated dysfunction, including our own work using ASO mediated PLN gene silencing, which remarkably led to a reversal in the R14Del heart failure phenotype in heterozygous R14del mouse model as well as a remarkable increase in viability in homozygous R14del mice.⁷ In addition, other reports including PLN ablation and PLN phosphomimic transfection underscore the importance of PLN as a heart failure target.⁸

The approach in our current report does not require altering the gene expression/replacement as we express a binding domain that interacts directly with the PLN protein. The use of antibody fragments allows specific targeting of particular conformational states (i. e. pentamers/monomers), PTM or specific mutants, including potentially the disease mutant R14del. This “surgical” approach to target the disease variant is much more challenging using genetic knock-down approaches and allows the inhibition of the disease variant without direct perturbation of target gene expression. Moreover, it is also known that PLN aggregation is a hallmark of PLNR14del pathology. Although the role of the aggregates in the pathology has yet to be determined, our approach may help to elucidate the role of these proteinaceous aggregates by specific targeting with particular intrabodies. Additionally, the intrabodies could further enable clearance of long-lived toxic aggregates that would be difficult to neutralise using conventional genetic knock-down strategies alone.

7. Grote Beverborg, N. *et al.* Phospholamban antisense oligonucleotides improve cardiac function in murine cardiomyopathy. *Nat Commun* **12**, 5180 (2021).

8. Iwanaga, Y. *et al.* Chronic phospholamban inhibition prevents progressive cardiac dysfunction and pathological remodeling after infarction in rats. *J Clin Invest* **113**, 727–736 (2004).

Reviewer #3 (Remarks to the Author): *My summary review is that the ms is very good, and breaks new ground. Methods and data analysis are sound, with sufficient detail for the antibody work. It is acceptable with minor modifications in the discussion. The dimerization to increase avidity and target the pentamer is especially impressive. The modRNA worked well for the very short-term experiments, although it is not clear that it is going to perform better than a DNA plasmid when the latter can be delivered efficiently in the chosen cell line in many circumstances. The discussion is already quite detailed. One issue that should also be considered – When delivering the intrabodies systemically, what is the chance of an off-target effect? Are the screening peptides shared with any other proteins? How tight is the control afforded by the promoter that was used with the AAV9, since AAV9 will also go to the brain?*

We whole-heartedly agree with Reviewer 3 that it is critical to have highly cardiac-specific expression of the intrabodies for clinical applications. Here, as a proof-of-concept *in vivo* study, we used a naturally cardiac tropic AAV9 capsid with a well-established heart-specific TNT455 promoter and achieved good intrabody expression in the heart. In addition to heart tissue samples, we have also

collected samples from liver and quadriceps of the MLP KO mice and assessed by western blotting if the intrabody was expressed in these tissues. We found no detectable expression in the two latter tissues, whereas expression in the heart was detected in all AAV9 B4B4 treated animals. We have added these new results in a revised figure 6 (and Supplementary Figure 9) and added the sentence on p 11, line 318: *“In addition, we also evaluated the expression levels of the transgene in liver tissue and quadriceps muscle taken from the same mice, showing no detectable expression of the VHH₂ B4B4 intrabody, emphasizing the tight control of the TNT 455 cardiac specific promoter (Fig 6 g; Supplementary Fig. 9 c,d).”*

Furthermore, when we delivered firefly luciferase by the same vector to mice via IV and visualized biodistributions of luciferase expression by IVIS imaging, we saw luciferase signals predominantly in the heart with marginal signals from the liver. We are happy to provide our internal confidential data if requested. Although AAV9 vector is known to transduce other tissues, including the brain, we didn't observe any notable luciferase expression apart from the heart and the liver, likely due to the use of the cardiac-specific TNT455 promoter. For future clinical applications, we can employ a next generation cardiac-specific AAV vector system with an engineered cardiac-targeted AAV capsid with reduced liver-tropism, along with a highly cardiac-specific promoter, to minimize off-targeted AAV transduction.

With respect to the antigen, phospholamban, we agree with the reviewer that it is conceivable that our intrabodies could cross-react with proteins, containing sequences similar to those contained in the PLN peptide epitope and thus potentially induce off-target effects. A proteomic analysis using immunoprecipitation and mass-spec analysis may identify possible interacting proteins. However, we did not find any significant adverse effects associated with the expression of the intrabodies in our cell models and *in vivo*.

To capture the points raised above, we have strengthened the discussion on p 13, line 380: *“Our study demonstrated the use of the less invasive and more homogenous, transduction via a naturally cardiotropic AAV9 vector under the control of a strong heart specific promoter for intrabody expression. This resulted in a prolonged expression of the intrabodies specifically in the heart. This tissue specificity minimises the risk of adverse effects arising from cross reactivity with other targets. This issue might be of particular relevance for intrabodies that recognise linear epitopes, such as our anti-PLN intrabodies, as these epitopes are likely to contain sequence motifs that are present in other proteins.”*

REVIEWERS' COMMENTS

Reviewer #1 (Remarks to the Author):

We thank the authors for replying adequately to our comments and adding new data to the manuscript. We think that the new information and corrections improved the manuscript.

There are a few points:

Since only calcium dynamics were measured and not sarcomere shortening, please change the text in this sentence:

1) The expression of the phospho-specific intrabodies VHH C6 and VHH2 C6C6 resulted in only minor effects on cardiomyocyte calcium kinetics e.g. the decay phase, leading to a more speculative model for the mode of action of these intrabodies.

Here, new experimental evidence would strongly increase the value of this MS.

2) In Line 1219, “?” should be replaced by “I”.

3) In Figure 2b and Figure 6f, the size of the PLN antibody is not the same compared to the marker, please unify them.

Reviewer #2 (Remarks to the Author):

The authors addressed all my concerns. I believe this paper is ready for publication in the current format.

Reviewer #3 (Remarks to the Author):

This ms is now significantly improved.

There is one sentence in the revised discussion that could use either no comma, or an additional comma, to make it more readable. ("Our study demonstrated the use of the less invasive and more homogenous, transduction via a naturally cardiotropic AAV9 vector under the control of a strong heart specific promotor for intrabody expression.")

POINT-BY-POINT RESPONSE TO REVIEWERS' COMMENTS

We are pleased and thank the Reviewers and Editors for their constructive criticism and for appreciating the improvements made during the revision. We have now implemented all requested editorial changes in the updated manuscript. Our revised manuscript including figures, as well as point-by-point rebuttal are provided for the reviewers and editors.

Reviewer #1 (Remarks to the Author):

We thank the authors for replying adequately to our comments and adding new data to the manuscript. We think that the new information and corrections improved the manuscript. We would like to thank the reviewer for the appreciation of our work and the additional data included during the revision. We are also grateful for the further insightful comments and suggestions.

There are a few points:

Since only calcium, dynamics were measured and not sarcomere shortening, please change the text in this sentence:

1) The expression of the phospho-specific intrabodies VHH C6 and VHH2 C6C6 resulted in only minor effects on cardiomyocyte calcium kinetics e.g. the decay phase, leading to a more speculative model for the mode of action of these intrabodies.

We would like to thank the reviewer for the suggestion, we have now modified our text accordingly.

Here, new experimental evidence would strongly increase the value of this MS.

2) In Line 1219, “?” should be replaced by “I”.

We would like to thank the reviewer for the diligent review of our manuscript, we have now made the adjustment.

3) In Figure 2b and Figure 6f, the size of the PLN antibody is not the same compared to the marker, please unify them.

We would like to thank the reviewer for the careful review, we have now made the adjustment.

Reviewer #2 (Remarks to the Author):

The authors addressed all my concerns. I believe this paper is ready for publication in the current format.

We would like to thank the reviewer for the appreciation of our work and the additional data included during the revision.

Reviewer #3 (Remarks to the Author):

This ms is now significantly improved.

There is one sentence in the revised discussion that could use either no comma, or an additional comma, to make it more readable. ("Our study demonstrated the use of the less invasive and more homogenous, transduction via a naturally cardiotropic AAV9 vector under the control of a strong heart specific promotor for intrabody expression.")

We would like to thank the reviewer for the appreciation of our work and the additional data included during the revision. We would like to thank the reviewer for the thorough review of our manuscript, we have now made the adjustment.